# Srs2 binding to proliferating cell nuclear antigen (PCNA) and its sumoylation contribute to replication protein A (RPA) antagonism during the DNA damage response

Jiayi Fan[1], Nalini Dhingra[1†], Tammy Yang[2], Vicki Yang[2], Xiaolan Zhao[1*]

[1]Molecular Biology Program, Memorial Sloan Kettering Cancer Center, New York, United States; [2]City University of New York Hunter College, New York, United States

*For correspondence:
zhaox1@mskcc.org

Present address: †Yale Center for Molecular Discovery, Yale University, West Haven, United States

Competing interest: The authors declare that no competing interests exist.

## eLife Assessment

This article reports **valuable** findings on the role of the Srs2 protein in turning off the DNA damage signaling response initiated by Mec1 (human ATR) kinase. The data provide **convincing** evidence that Srs2 interaction with PCNA and ensuing SUMO modification is required for checkpoint down-regulation. However, while the model that Srs2 acts at gaps after camptothecin-induced DNA damage is reasonable, direct experimental evidence for this is currently lacking. The work will be of interest to cell biologists studying genome integrity.

**Abstract** Activation of the DNA damage checkpoint upon genotoxin treatment induces a multitude of cellular changes to cope with genome stress. After prolonged genotoxin treatment, the checkpoint can be downregulated to allow cell cycle and growth resumption. In yeast, down-regulation of the DNA damage checkpoint requires the Srs2 DNA helicase, which removes the ssDNA binding complex replication protein A (RPA) and the associated Mec1 checkpoint kinase from DNA, thus dampening Mec1-mediated checkpoint. However, it is unclear whether the 'anti-checkpoint' role of Srs2 is temporally and spatially regulated to allow timely checkpoint termination while preventing superfluous RPA removal. Here we address this question by examining regulatory elements of Srs2, such as its phosphorylation, sumoylation, and protein-interaction sites. Our genetic analyses and checkpoint level assessment suggest that the RPA countering role of Srs2 is promoted by Srs2 binding to proliferating cell nuclear antigen (PCNA), which recruits Srs2 to a subset of ssDNA containing regions. RPA antagonism is further fostered by Srs2 sumoylation, which we found depends on the Srs2-PCNA interaction and Mec1, and peaks after Mec1 activity reaches maximal levels. These data support a model in which Srs2 recruitment to PCNA adjacent to ssDNA-RPA filaments, followed by Mec1-dependent sumoylation, modulates RPA-mediated checkpoint signaling, while Srs2 action is limited at ssDNA regions lacking proximal PCNA, thereby favoring RPA-mediated ssDNA protection and repair.

## Introduction

The DNA damage response promotes organismal survival of genotoxic stress by eliciting many cellular changes. A critical component of this response is a signaling process termed the DNA damage checkpoint (DDC) (*Enoch et al., 1992*; *Hartwell and Weinert, 1989*; *Lanz et al., 2019*). Upon experiencing

genotoxic stress, apical DDC kinases are recruited to DNA lesion sites by the damage sensing proteins and are then activated by activator proteins (*Usui et al., 2001*; *Zou and Elledge, 2003*). A universal DDC sensor is the replication protein A (RPA) complex that binds with high affinity and specificity to single-strand DNA (ssDNA), whose abundance rises under almost all types of genotoxic stress (*Maréchal and Zou, 2015*). Using the budding yeast system as an example, RPA bound to the ssDNA filament can interact with Ddc2 (homolog of the human ATRIP protein), the co-factor of the Mec1 (homolog of the human ATR protein) checkpoint kinase, and recruit the Mec1-Ddc2 complex to DNA lesion sites (*Elledge, 1996*; *Foiani et al., 2000*). Subsequently, Mec1 is activated by activator proteins, such as the 9-1-1 complex loaded at the 5' DNA ends at ssDNA and double strand (ds) DNA junctions. Activated Mec1 can then phosphorylate multiple substrates, among which is a key downstream effector kinase Rad53, the phosphorylation of which turns on its own kinase activity and yields further phosphorylation events (*Harrison and Haber, 2006*). Collectively, the DDC kinases can modify hundreds of proteins, inducing a multitude of pathway changes (*Lanz et al., 2019*). For example, DDC can arrest or delay the cell cycle progression to allow more time for genome repair (*Sanchez et al., 1999*; *Wang et al., 2001*).

While DDC activation is crucial for the cell to cope with genotoxic stress, its timely downregulation is equally vital (*Waterman et al., 2020*). Many changes induced by the DDC kinases need to be reversed to allow continued cell growth and recovery. For example, the DDC-mediated cell cycle arrest has to be lifted to allow proliferation and alternative DNA repair mechanisms permitted at other cell cycle phases (*Pellicioli et al., 2001*; *Waterman et al., 2020*). Thus far, DDC downregulation has been mainly studied using the model organism yeast, with several strategies being identified (*Waterman et al., 2020*). One of these is mediated by the DNA helicase Srs2 that targets the most upstream step of the DDC pathway through removing RPA and the associated Mec1-Ddc2 from chromatin (*Dhingra et al., 2021*). Importantly, the role of Srs2 in DDC dampening (anti-checkpoint) can be separated from its well-studied role in removing the recombinase Rad51 from ssDNA (anti-recombination) that modulates homologous recombination levels and outcomes (*Dhingra et al., 2021*). The former role of Srs2 is critical for cell survival of genotoxins, such as camptothecin (CPT) that traps Top1 on DNA (*Dhingra et al., 2021*). Considering that other organisms also possess DNA helicases capable of removing RPA from ssDNA (*Hormeno et al., 2022*; *Jenkins, 2021*; *Shorrocks et al., 2021*), it is plausible that an anti-checkpoint strategy involving RPA antagonism may be present beyond yeast.

Currently, little is known about how checkpoint dampening can be temporally controlled to minimize interference with the initial DDC activation that must occur in the early phase of the DNA damage response. In the case of the Srs2-mediated checkpoint regulation, it is also unclear how the process can minimize superfluous removal of RPA from ssDNA sites that are unlikely to induce the DDC due to lacking 5' DNA ends at ss-dsDNA junctions for 9-1-1 loading, such as at those formed at R-loops. The main functions of RPA at these sites can be ssDNA protection and repair, rather than DDC induction, thus these sites may benefit from shielding from Srs2-mediated RPA removal. In addressing these questions, we hypothesize that Srs2-based RPA antagonism should be selective in location and timing, and that the regulatory elements within the Srs2 protein may contribute to such selectivity.

To examine the above hypothesis and to define Srs2 features relevant to its RPA antagonism during DDC regulation, we inspected Srs2 protein sites involved in its post-translational modifications and interactions with other proteins. Our genetic analyses of mutations affecting each of these sites suggest that while most of the examined features do not affect RPA antagonism, Srs2 binding to PCNA, which marks 3' DNA ends of ss-dsDNA junctions that flank ssDNA gaps or tails, and Srs2 sumoylation contribute to RPA regulation in the context of DDC. Srs2 binding to PCNA is known to recruit Srs2 to replication perturbation sites, while the roles of Srs2 sumoylation were less clear (*Kolesar et al., 2012*; *Moldovan et al., 2007*; *Papouli et al., 2005*; *Pfander et al., 2005*; *Saponaro et al., 2010*). Interestingly, we found that Srs2 binding to PCNA and Srs2 sumoylation are functionally related since PCNA binding promotes Srs2 sumoylation. Moreover, Srs2 sumoylation level peaks late in the response to genomic stress after the DDC kinases are hyper-activated, and this modification depends on the Mec1 checkpoint kinase. Collectively, our data suggest a model wherein Srs2 can distinguish between two types of ssDNA regions using PCNA proximity as a guide for subsequent sumoylation, RPA removal, and countering hyper-activated checkpoint.

## Results

### Srs2 regulatory features and their disabling mutants

To understand which Srs2 attributes are relevant to RPA regulation during the DNA damage response, we examined mutant alleles that perturb known features of the Srs2 protein. Srs2 was reported to contain a helicase domain at its N-terminal region and three protein–protein interaction sites at its C-terminal region (*Figure 1A*; *Niu and Klein, 2017*). We have previously shown that the Srs2 ATPase-dead allele (*srs2-K41A*) fails to downregulate both RPA levels on chromatin and the Mec1-mediated DDC, indicating a requirement of ATPase-driven stripping of RPA from ssDNA (*Dhingra et al., 2021*). The reported Srs2 protein-binding sites include a Rad51 binding domain (Rad51-BD), a PCNA inter-action motif (PIM), and a SUMO interaction motif (SIM) (*Figure 1A*; *Armstrong et al., 2012*; *Colavito et al., 2009*; *Kolesar et al., 2016*; *Kolesar et al., 2012*; *Papouli et al., 2005*). We employed CRISPR-based editing to generate the *srs2-ΔPIM* (Δ1159–1163 amino acids) and the *srs2-SIM^{mut}* ($_{1170}$IIVID$_{1174}$ to $_{1170}$AAAAD$_{1174}$) alleles used in previous studies and examined an established *srs2-ΔRad51BD* allele (Δ875–902 amino acids) (*Armstrong et al., 2012*; *Colavito et al., 2009*; *Kolesar et al., 2016*; *Kolesar et al., 2012*; *Papouli et al., 2005*). We note that despite that Srs2 can remove RPA from ssDNA, an interaction between the two proteins has yet to be reported.

We also queried post-translational modifications reported for Srs2 by mutating the corresponding modification sites (*Figure 1A*). First, seven Cdk1 phosphorylation consensus sites of Srs2 were mutated to produce the *srs2-7AV* (T604V, S698A, S879A, S893A, S938A, S950A, and S965A) allele (*Chiolo et al., 2005*), which was previously shown to affect homologous recombination. Second, three sumoy-lation sites of Srs2 were mutated to generate *srs2-3KR* (K1081R, K1089R, and K1142R) (*Saponaro et al., 2010*). Finally, we mutated two serine residues (S890 and S933) to alanine to generate *srs2-2SA*, since phosphorylation at these sites was identified in proteomic studies as dependent on both the Mec1 kinase and genotoxic treatments (*Faca et al., 2020*). Because the Srs2 C-terminal end contrib-utes to the protein's functions (*Antony et al., 2009*; *Colavito et al., 2009*), the six mutant alleles described above were not tagged with epitopes at its C-terminus. Using an anti-Srs2 specific anti-body, we showed that all six alleles supported wild-type levels of Srs2 protein based on immunoblot-ting (*Figure 1B*).

### Mutating PCNA binding and sumoylation sites of Srs2 rescues *rfa1-zm2* CPT sensitivity

Srs2 loss leads to persistent DDC and perturbed recombination, both of which contribute to *srs2Δ* sensitivity to genotoxins, including CPT and methyl methanesulfonate (MMS) (*Figure 2A*; *Niu and Klein, 2017*). Significantly, our recent study has shown that RPA mutants that reduce its ssDNA binding affinity rescue the persistent DDC levels in *srs2Δ* cells but not their recombination defects (*Dhingra et al., 2021*). *rfa1-zm2* (N492D, K493R, K494R) is one of the RPA alleles examined that contains mutations at three DNA contacting residues in the Rfa1 subunit of RPA (*Fan and Pavletich, 2012*). Purified RPA complex containing these mutations exhibited a twofold increase in dissociation constant using ssDNA as substrate, compared to the wild-type RPA complex (*Dhingra et al., 2021*). In cells, *rfa1-zm2* supports normal Rfa1 protein level, cell growth, DNA replication (*Dhingra et al., 2021*). It also did not affect gene conversion, direct repeat repair or rDNA recombination, nor the hyper-recombination phenotype of *srs2Δ* (*Dhingra et al., 2021*). However, *rfa1-zm2* and *srs2Δ* are mutually suppressive for both their genotoxic sensitivities and their DDC abnormalities (*Figure 2B*, *Figure 2—figure supplement 1A*; *Dhingra et al., 2021*). These data suggest that the *srs2* and *rfa1-zm2* genetic relationship can be used as a readout for the antagonism between Srs2 and RPA in DDC and that this functional relationship can be separable for Srs2's anti-recombination role stemming from Rad51 regulation. Based on this rationale, we employed the genetic suppression as an initial readout in querying whether any of the six *srs2* mutant alleles described above are related to RPA regulation. Given that Srs2's role in checkpoint dampening was best examined in CPT (*Dhingra et al., 2021*), which induces the DNA damage checkpoint to delay G2/M progression, but does not induce the DNA replication checkpoint, experiments were mostly carried out in this condition.

We found that on their own, *srs2 -ΔPIM*, -*SIM^{mut}*, and −*7AV* showed different levels of sensitivity toward CPT at a concentration ranging from 2 to 8 µg/ml, while *srs2-ΔRad51BD* and -*3KR* exhib-ited close to wild-type level of CPT resistance, which is consistent with previous reports (*Colavito et al., 2009*; *Saponaro et al., 2010*; *Figure 2—figure supplement 1A*). In addition, *srs2-2SA*, which

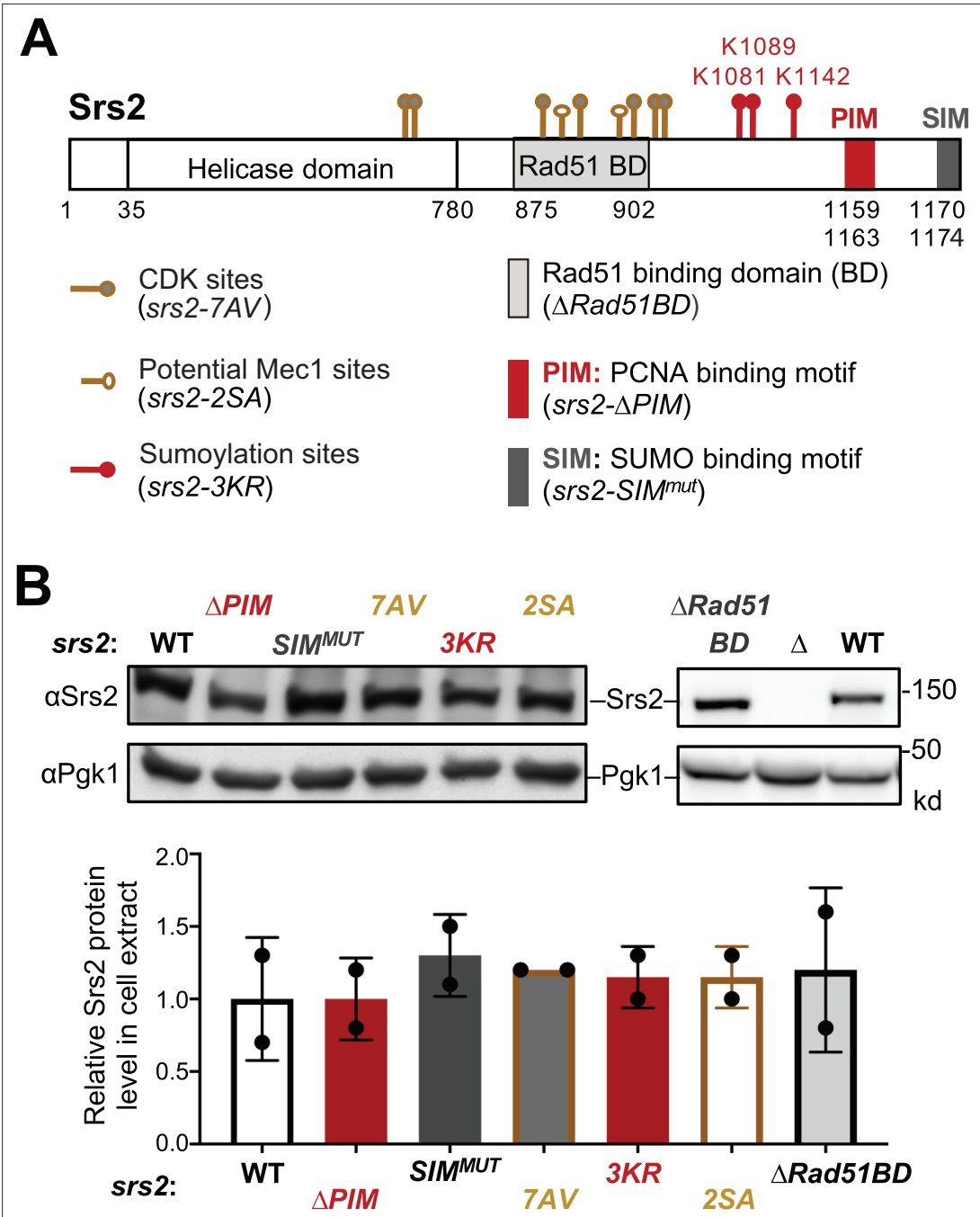

**Figure 1.** Regulatory features of Srs2 and corresponding mutants. (**A**) Schematic of Srs2 protein domains, regulatory features, and mutations affecting each of its features. Cdk1 phosphorylation sites depicted here include T604, S698, S879, S893, S938, S950, and S965 (**Chiolo et al., 2005**). Potential Mec1 phosphorylation sites include S890 and S933 (**Albuquerque et al., 2015**; **Faca et al., 2020**). Sumoylation sites mapped previously include K1081, K1089, and K1142 (**Saponaro et al., 2010**). Protein-interaction domains or motifs include the Rad51 binding domain (BD), PCNA Interaction Motif (PIM), and SUMO Interaction Motif (SIM) (**Colavito et al., 2009**; **Kolesar et al., 2016**; **Kolesar et al., 2012**). The residues for these domains are indicated. Mutant alleles disabling each of these features are included in the parentheses. (**B**) Mutations affecting Srs2 features do not affect Srs2 protein level. Protein extracts were prepared from asynchronous cells. Srs2 was examined using anti-Srs2 antibody by immunoblotting. Srs2 levels in each mutant were first normalized to the Pgk1 loading control and then to those in wild-type cells. Mean of two biological isolates per genotype is graphed with error bars representing standard deviation (SD). No statistical differences were seen when comparing the Srs2 protein level between the wild-type (WT) control and each of the indicated mutants based on Student's *t*-test.

The online version of this article includes the following source data for figure 1:

**Source data 1.** Original western blots for **Figure 1B**, indicating the relevant lanes.

**Source data 2.** Original files for western blot analysis displayed in **Figure 1B**.

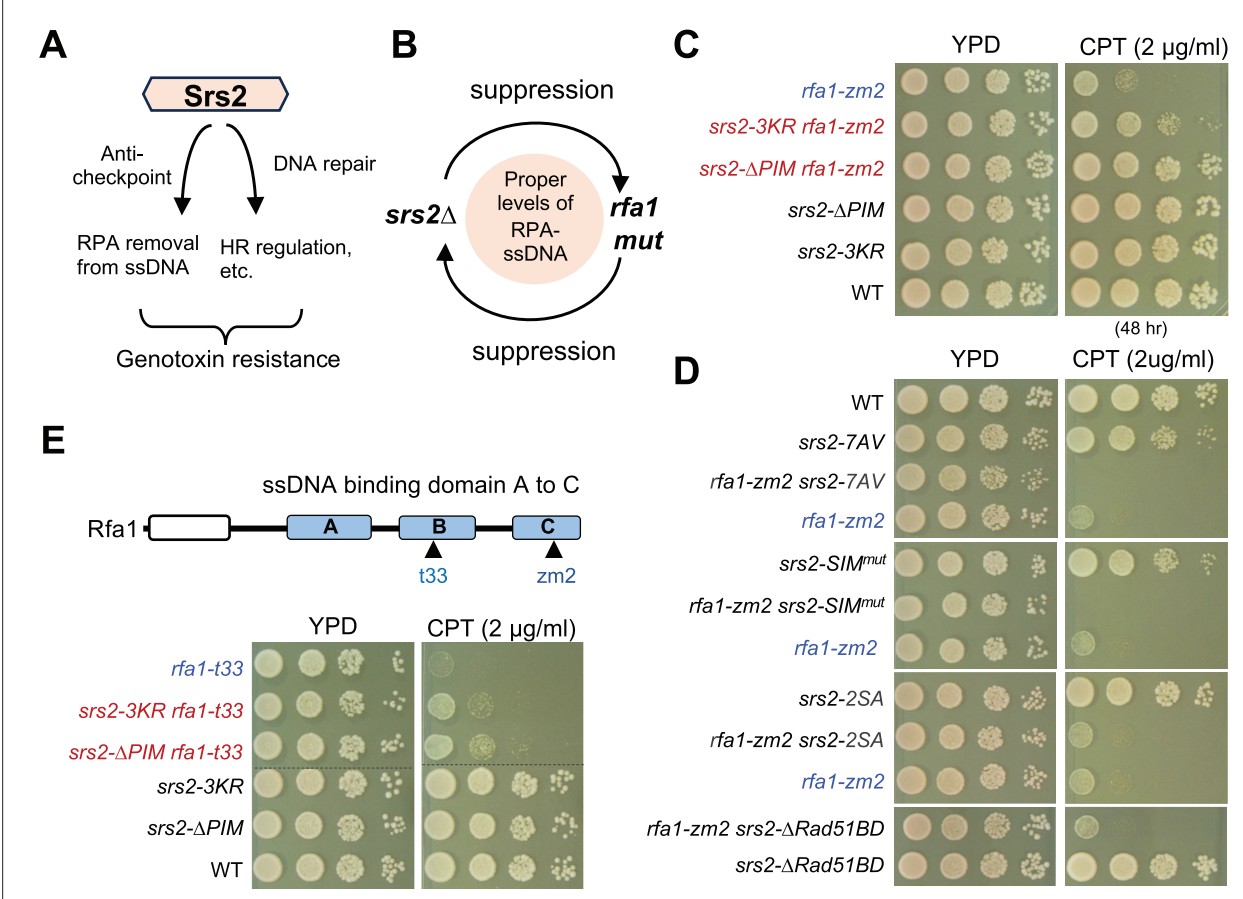

**Figure 2.** PCNA binding and sumoylation are involved in suppressing *rfa1* mutants. (**A**) Schematic to show that Srs2's roles in checkpoint dampening (anti-checkpoint for simplicity) and DNA repair, both of which contribute to genotoxin resistance. (**B**) Schematic to highlight the mutual suppressive relationship between *srs2Δ* and *rfa1* mutants and the underlying effects on the RPA-ssDNA levels in cells. (**C**) *srs2-ΔPIM* or *-3KR* suppresses *rfa1-zm2* sensitivity to CPT. A 10-fold serial dilution of cells of the indicated genotypes was spotted and growth was assessed after incubation at 30°C for 40 hours unless otherwise noted. (**D**) Several *srs2* mutants did not suppress *rfa1-zm2* sensitivity to CPT. Experiments were performed as in (**C**). (**E**) *srs2-ΔPIM* or *-3KR* suppresses *rfa1-t33* sensitivity to CPT. Top: schematic of the Rfa1 protein domains and *rfa1* mutant alleles. Bottom: *srs2-ΔPIM* or *-3KR* improved CPT sensitivity of *rfa1-t33* cells. Experiments were performed as in (**C**).

The online version of this article includes the following figure supplement(s) for figure 2:

**Figure supplement 1.** *srs2* mutants examined for CPT sensitivity and interactions with *rfa1* mutant.

has not been examined previously, did not show CPT sensitivity (*Figure 2—figure supplement 1A*). Since *rfa1-zm2* cells exhibited stronger CPT sensitivity than the tested *srs2* alleles (*Figure 2—figure supplement 1A*), we asked if the latter could suppress *rfa1-zm2* CPT sensitivity. We found that both *srs2-ΔPIM* and *-3KR* improved the CPT resistance of *rfa1-zm2* cells on media containing 2 μg/ml CPT (*Figure 2C*). Rescue was also seen at higher CPT concentrations, with *srs2-ΔPIM* exhibiting a stronger effect than *srs2-3KR* (*Figure 2—figure supplement 1B*). In contrast to *srs2-ΔPIM* and *srs2-3KR*, *srs2-ΔRad51BD* and *-2SA* showed no effect on *rfa1-zm2* growth on media containing CPT, whereas *srs2-7AV and -SIM^mut* worsened it (*Figure 2D*). The differential effects seen for the *srs2* alleles are consistent with the notion that Srs2 has multiple roles in cellular survival of genotoxic stress (*Bronstein et al., 2018*; *Niu and Klein, 2017*). The unique suppression seen for mutants affecting Srs2 binding to PCNA and its sumoylation suggests that these features are relevant to RPA regulation during the DNA damage response.

## Additional suppressive effects conferred by *srs2-ΔPIM* and *-3KR* toward *rfa1* mutants

We next tested whether the observed *srs2-ΔPIM* and *-3KR* suppression was specific towards *rfa1-zm2* or applicable to other *rfa1* mutants that are also defective in ssDNA binding. While *rfa1-zm2* affects the ssDNA binding domain C (DBD-C) of Rfa1, a widely used *rfa1* allele, *rfa1-t33*, impairs the DBD-B domain (*Figure 2E*; *Umezu et al., 1998*). Unlike *rfa1-zm2*, *rfa1-t33* additionally decreases Rfa1 protein level and interferes with DNA replication and repair (*Deng et al., 2014*; *Umezu et al., 1998*). Despite having these additional defects, CPT survival of *rfa1-t33* cells was also improved by *srs2-ΔPIM* or *srs2-3KR* (*Figure 2E*). As seen in the case of *rfa1-zm2*, *srs2-ΔPIM* conferred better rescue than *srs2-3KR* toward *rfa1-t33* (*Figure 2E*). When comparing the two *rfa1* alleles, *srs2* rescue was stronger toward *rfa1-zm2*, likely because *rfa1-t33* has other defects besides reduced ssDNA binding (*Figure 2C and E*). In either case, *srs2* mutants only conferred partial suppression, consistent with the notion that Srs2 has roles beyond RPA regulation, such as in Rad51 regulation and removing DNA joint molecules (*Bronstein et al., 2018*; *Niu and Klein, 2017*). Nevertheless, the observed rescue of both *rfa1-zm2* and *rfa1-t33* provided strong genetic evidence that Srs2 binding to PCNA and its sumoylation are involved in the regulation of RPA during the DNA damage response.

Srs2 regulation of RPA during DDC is not only observed under CPT treatment, but is also seen in treatment with MMS (*Dhingra et al., 2021*), which induces both DDC and the DNA replication checkpoint, thus delaying both S phase progression and G2/M phase exit (*Paciotti et al., 2000*; *Tercero et al., 2003*). Consistent with observation made in CPT conditions, *srs2-ΔPIM* moderately improved the MMS sensitivity of *rfa1-zm2* and *rfa1-t33*, while *srs2-3KR* had a small effect on the MMS sensitivity of *rfa1-zm2* (*Figure 2—figure supplement 1B and C*). The degrees of suppression seen in MMS treatment were less than those observed in CPT conditions, and we address likely reasons for this difference in Discussion. Collectively, the multiple genetic results are consistent among themselves and provide evidence that the roles of Srs2 binding to PCNA and sumoylation in RPA antagonism can be applicable to more than one genomic stress situation.

## *srs2-ΔPIM* and *-3KR* rescue checkpoint abnormality of *rfa1-zm2* cells

We next investigated whether *srs2-ΔPIM* and *-3KR* suppression of the DNA damage sensitivity of *rfa1-zm2* cells pertained to DDC regulation. We employed two readouts for DDC that are widely used. First, we used the F9 antibody, which specifically recognizes phosphorylated Rad53, to measure the levels of activated Rad53 (*Fiorani et al., 2008*). Second, we measured cellular ability to timely exit the G2/M phase after CPT treatment using FACS analyses. We note that CPT-induced DDC leads to G2/M delay rather than an arrest as shown in previous studies (*Dhingra et al., 2021*; *Menin et al., 2018*; *Redon et al., 2003*). We recently reported that *srs2Δ* cells exhibit increased levels of phosphorylated Rad53 and diminished ability to timely exit G2/M phase in CPT, and that both are suppressed by *rfa1-zm2* (*Dhingra et al., 2021*). Interestingly, *rfa1-zm2* leads to similar defects and these are rescued by *srs2Δ* (*Dhingra et al., 2021*; *Figure 3A*). The mutual suppression seen in these assays is consistent with that seen for CPT sensitivity (*Dhingra et al., 2021*). We note that several other RPA ssDNA binding mutants, such as *rfa1-t33*, also exhibit phenotypes indicative of increased DDC levels (*Pellicioli et al., 2001*; *Seeber et al., 2016*; *Smith and Rothstein, 1995*; *Zou and Elledge, 2003*). This is likely because compromised ssDNA protection can lead to increased levels of DNA lesions, thus enhancing the activation of Mec1 and/or its homolog Tel1.

We reasoned that if *srs2-ΔPIM* and *srs2-3KR* compromise Srs2's ability to remove RPA from DNA in cells, they should alleviate *rfa1-zm2* cells' defects in ssDNA protection and the associated DDC abnormalities (*Figure 3A*). Indeed, while *rfa1-zm2* exhibited about a fivefold increase in Rad53 phosphorylation levels in CPT compared with wild-type cells as seen before (*Dhingra et al., 2021*), *srs2-ΔPIM* and *-3KR* reduced this level by about 50% and 80%, respectively (*Figure 3B*). Similar to earlier data, we observed that wild-type cells arrested in $G_2$/M after 1 hour of CPT treatment and about 25% of these cells entered $G_1$ after another hour (*Figure 3C*; *Dhingra et al., 2021*). The ability to timely exit from $G_2$/M was reduced in *rfa1-zm2* cells so that only 13% cells moved to G1 after 2-hour treatment with CPT, as seen before (*Figure 3C*; *Dhingra et al., 2021*). Compared with *rfa1-zm2* cells, *srs2-ΔPIM rfa1-zm2* and *srs2-3KR rfa1-zm2* cells showed increased percentages of cells transitioning into the G1 phase (*Figure 3C*). The suppressive effects seen in both assays were stronger for *srs2-ΔPIM* than *srs2-3KR* as seen in genotoxicity assays (*Figure 2—figure supplement 1A*; *Figure 3B and C*). When

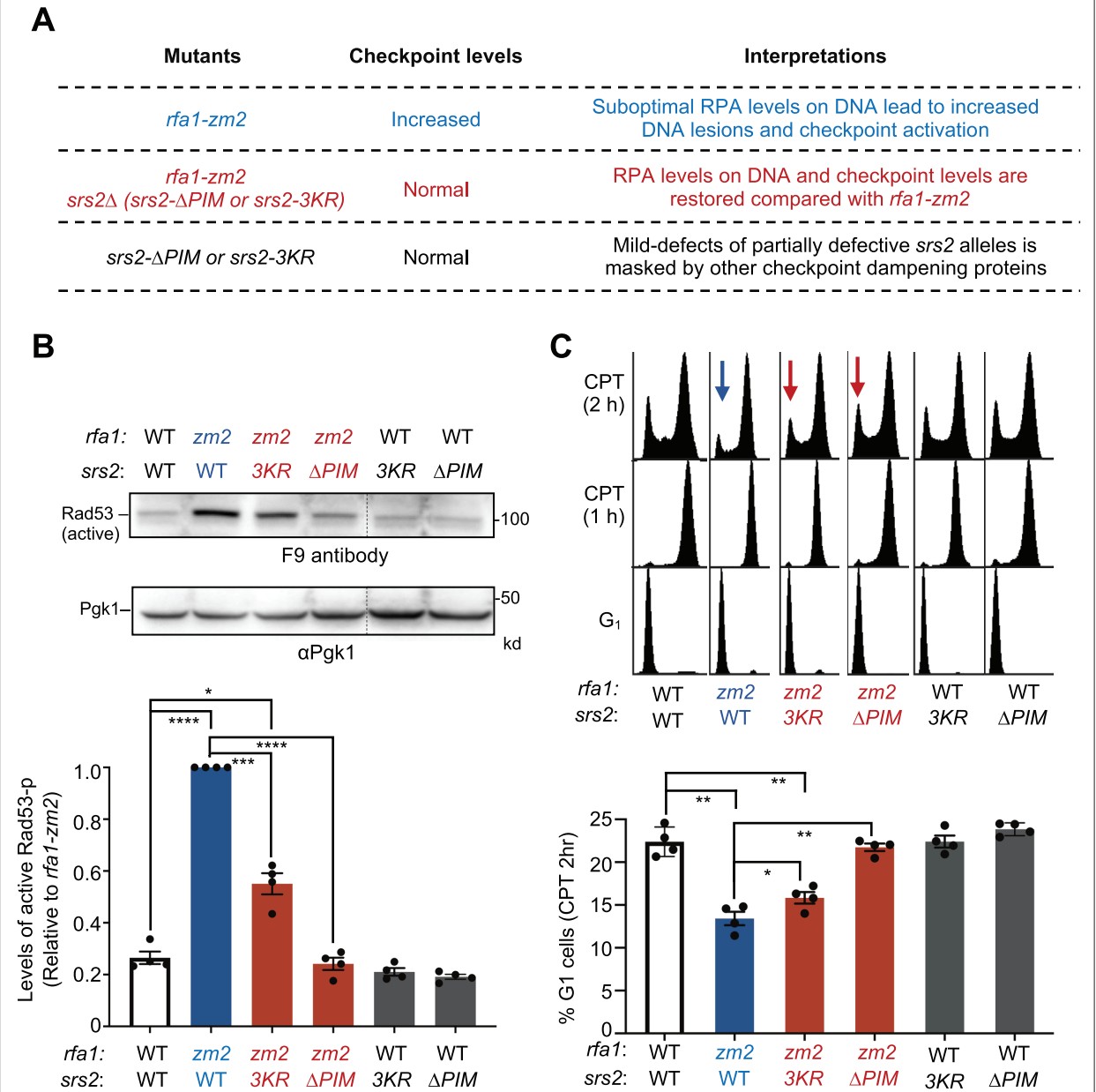

**Figure 3.** *PIM* and *-3KR* correct checkpoint abnormalities in *rfa1-zm2* cells. (**A**) A table summarizing DDC levels observed in *rfa1-zm2*, *srs2* null or partially defective mutants, and their combined mutants in this work and in a previous study (***Dhingra et al., 2021***) and interpretations. (**B**) *srs2-ΔPIM* or *-3KR* reduces the levels of phosphorylated Rad53 in *rfa1-zm2* cells. Protein extracts were examined after G1 cells were released into cycling in the presence of 16 ug/ml CPT for 2 hours. Phosphorylated Rad53 (active form) was detected by the F9 antibody by immunoblotting. Phosphorylated Rad53 signals in each indicated strain were compared to the Pgk1 loading control and normalized to those of *rfa1-zm2*. Two biological isolates per genotype were examined in two technical repeats, and results are graphed with error bars representing SD. Statistical analysis was performed using pairwise Student's *t*-test (*p<0.05; ***p<0.001; ****p<0.0001, and non-statistical significance is not marked) (**C**) *srs2-ΔPIM* and *-3KR* allow better G1 entry of *rfa1-zm2* cells. Experiments were performed as in (**A**), and samples were collected at indicated time points. FACS profiles of the samples are shown at the top and percentages of G1 cells after 2 hours of CPT treatment (CPT 2 h) are plotted at the bottom. Two biological isolates per genotype were examined in two technical repeats and results are graphed with error bars representing SD. Statistical analysis was performed and results are depicted as in (**B**).

The online version of this article includes the following source data and figure supplement(s) for figure 3:

**Source data 1.** Original western blots for ***Figure 3B***, indicating the relevant lanes.

**Source data 2.** Original files for western blot analysis displayed in ***Figure 3B***.

**Figure supplement 1.** *srs2-ΔPIM* cells exhibit increased levels of Rfa1 on chromatin.

*Figure 3 continued on next page*

*Figure 3 continued*

**Figure supplement 1—source data 1.** Original western blots for *Figure 3—figure supplement 1*, indicating the relevant lanes.

**Figure supplement 1—source data 2.** Original files for western blot analysis displayed in *Figure 3—figure supplement 1*.

assaying for chromatin RPA levels, we found that while *srs2-Δpim* showed an increase compared with the wild-type control, *rfa1-zm2* reversed this effect (*Figure 3—figure supplement 1*). This data supports the Srs2 and RPA antagonism. Collectively, results from the above assays are consistent with each other and suggest that Srs2 binding to PCNA and its sumoylation contribute to RPA antagonism during DDC regulation.

## *srs2-ΔPIM* is additive with an *slx4* mutant defective in DDC recovery

We found that *srs2-ΔPIM* and *srs2-3KR* mutants on their own behaved normally in the two DDC assays described above, which differs from *srs2Δ* cells that exhibit increased levels of Rad53 phosphorylation and reduced G1 entry (*Figure 3B and C*; *Dhingra et al., 2021*). One plausible interpretation of this difference is that unlike *srs2Δ*, these partially defective *srs2* mutants only mildly impair Srs2-mediated RPA regulation during DDC downregulation, such that their defects can be compensated for by other DDC dampening systems. To test this idea, we examined the consequences of removing the Slx4-mediated anti-checkpoint function in *srs2-ΔPIM* cells. Slx4 is known to bind the scaffold protein Rtt107, and their complex can compete with the DDC protein Rad9 for binding to damaged chromatin, consequently disfavoring Rad9's role in promoting Rad53 phosphorylation (*Ohouo et al., 2010*; *Ohouo et al., 2013*). Slx4 uses an Rtt107 interaction motif (RIM) to bind Rtt107 and a previously established RIM mutant, *slx4^RIM* (T423A, T424A, and S567A), abolishes Rtt107 binding without affecting other Slx4 protein attributes (*Wan et al., 2019*). *slx4^RIM* was tagged with a TAP tag at its C-terminal end and examined together with a control wherein the wild-type Slx4 was tagged in the same manner. Compared with the control, *slx4^RIM* exhibited mild sensitivity toward CPT (*Figure 4A*). Significantly, when *srs2-ΔPIM* was combined with *slx4^RIM*, the double mutant exhibited stronger CPT sensitivity compared with corresponding single mutants (*Figure 4A*).

We further examined DDC levels when cells were exposed to CPT using both Rad53 phosphorylation and ability to timely exit G2/M as readouts. Compared with *SLX4-TAP* cells, *slx4^RIM-TAP* cells exhibited a twofold increase in Rad53 phosphorylation levels, consistent with a role of Slx4 in downregulating DDC (*Figure 4B*). In the *SLX4-TAP* background, *srs2-ΔPIM* led to a twofold increase in Rad53 phosphorylation level (*Figure 4B*). As cells containing untagged Slx4 and *srs2-ΔPIM* did not show Rad53 phosphorylation changes (*Figure 3B*), tagging Slx4 appeared to sensitize *srs2-ΔPIM*. Significantly, the *srs2-ΔPIM slx4^RIM-TAP* double mutant exhibited further increase in Rad53 phosphorylation than either *srs2-ΔPIM* or *slx4^RIM-TAP* single mutant (*Figure 4B*). Congruous with this finding, the *srs2-ΔPIM slx4^RIM-TAP* double mutant also reduced the percentage of cells entering $G_1$ compared with their single mutants after 2-hour treatment with CPT (*Figure 4C*). Together, these three lines of evidence support the notion that defects in the Srs2-mediated DDC downregulation can be compensated for by Slx4-mediated mechanisms.

## Srs2 binding to PCNA facilitates Srs2 sumoylation

We next addressed whether the two Srs2 features involved in RPA antagonism are functionally related. To this end, we tested if Srs2 binding to PCNA affected Srs2 sumoylation in cells. We confirmed that endogenous sumoylation of Srs2 could be detected in cell extracts after enriching SUMOylated proteins using 8His-tagged SUMO (8His-Smt3) (*Figure 5A*). Briefly, proteins were extracted from cells under denaturing conditions to disrupt protein–protein association and prevent desumoylation, and sumoylated proteins were enriched on nickel-nitrilotriacetic acid (Ni-NTA) resin (*Ulrich and Davies, 2009*). Consistent with previous findings, sumoylated forms of Srs2 in the Ni-NTA eluate were detected on immunoblots using an anti-Srs2 antibody only when cells contained 8His-Smt3, where unmodified Srs2 was seen in the presence or absence of 8His-Smt3 due to nonspecific binding to the resin (*Figure 5—figure supplement 1*; *Saponaro et al., 2010*). These bands were only seen in wild-type cells but not *srs2Δ* cells (*Figure 5—figure supplement 1*). We found that Srs2 sumoylation level was reduced in *srs2-3KR* strains in CPT conditions, as expected from mutating the three major sumoylation sites (*Figure 5A*; *Saponaro et al., 2010*). Interestingly, *srs2-ΔPIM* cells exhibited a similar

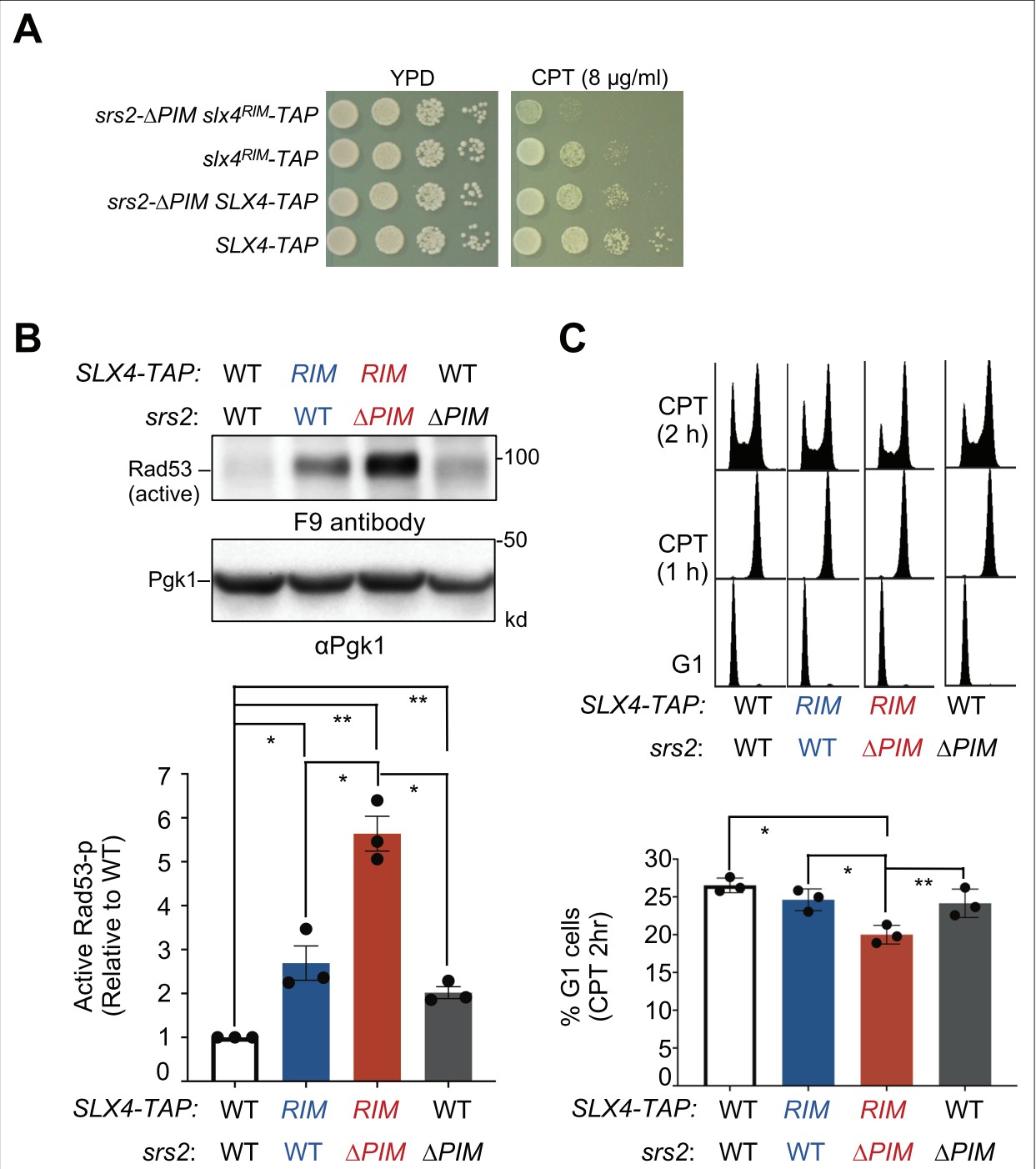

**Figure 4.** *srs2-ΔPIM* and *slx4^RIM* are additively defective in DDC downregulation. (**A**) The *srs2-ΔPIM slx4^RIM* double mutant shows stronger CPT sensitivity than either single mutant. Experiments are performed as in *Figure 2C*. (**B**) *srs2-ΔPIM* and *slx4^RIM* are additive for increasing the level of phosphorylated Rad53 when cells are treated with CPT. Experiments were conducted, and data are quantified as those described in *Figure 3B*. Results from three biological isolates per genotype are graphed with error bars representing SD. Statistical analysis was performed using pairwise Student's *t*-test. *p<0.05; **p<0.01. (**C**) *srs2-ΔPIM* and *slx4^RIM* are additive for reducing cells exiting from G2/M phase at 2 hours post CPT treatment. Experiments were conducted and data are quantified as those described in *Figure 3C*. Results from three biological isolates per genotype are graphed with error bars representing SD. Statistical analysis was performed using pairwise Student's *t*-test. *p<0.05; **p<0.01.

The online version of this article includes the following source data for figure 4:

**Source data 1.** Original western blots for *Figure 4B*, indicating the relevant lanes.

**Source data 2.** Original files for western blot analysis displayed in *Figure 4B*.

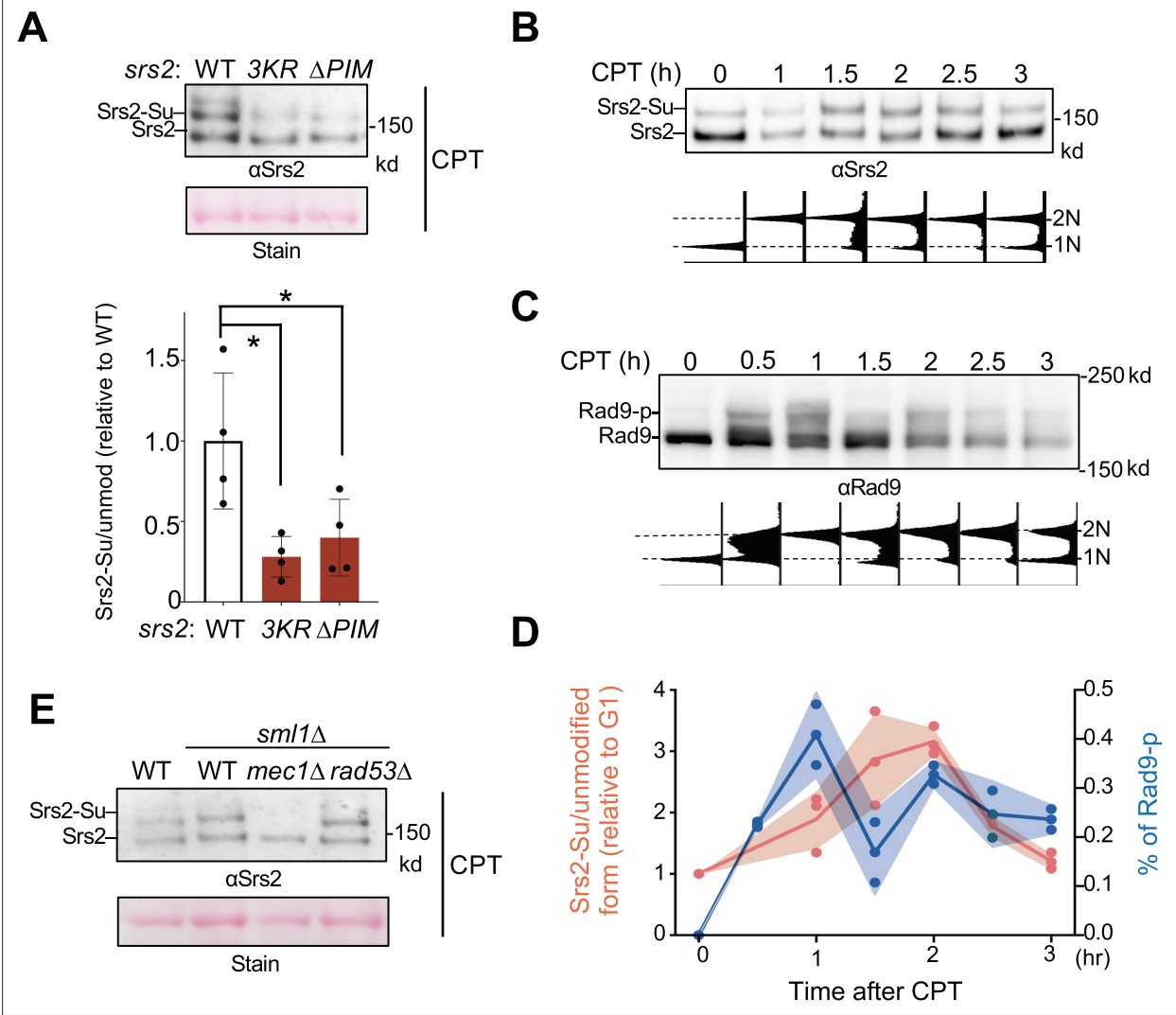

**Figure 5.** Sumoylation depends on the Srs2 PIM motif and the Mec1 kinase. (**A**) Srs2 sumoylation level is reduced in *srs2-ΔPIM* cells. Samples were collected after CPT treatment as in *Figure 3A* and sumoylated proteins were enriched as described in the text and 'Materials and methods'. Sumoylated form of Srs2 (Srs2-Su) migrated slower than unmodified Srs2 on gel and both were detected using anti-Srs2 antibody in immunoblotting. Equal loading is indicated by Ponceau-S stain (Stain). The levels of sumoylated Srs2 relative to those of unmodified Srs2 were normalized to wild-type cells and plotted. Two biological isolates per genotype were examined in two technical repeats and results are graphed with error bars representing SD. Statistical analysis was performed using pairwise Student's *t*-test. *$p < 0.05$. (**B**) Srs2 sumoylation level during CPT treatment. Samples were collected as described in *Figure 3B*. Experiments were conducted, and immunoblotting data is presented as those in (**A**). Time 0 samples are G1 cells before CPT treatment. FACS analysis at corresponding time points is included below the plot. (**C**) Rad9 phosphorylation level during CPT treatment. Samples used in (**B**) were examined for Rad9 phosphorylation using an anti-Rad9 antibody. Unmodified and phosphorylated (Rad9-p) Rad9 forms are indicated. FACS analysis at corresponding time points is indicated below blots. (**D**) Quantification of Srs2 sumoylation and Rad9 phosphorylation level changes during CPT treatment. Srs2 sumoylation signals were normalized based on unmodified form. Values at each timepoint were relative to that of G1 sample (Time 0) before CPT treatment, and fold changes were indicated on left y-axis. Rad9 signals were normalized to non-modified form and percentage of phosphorylated form was indicated on right y-axis. Results from three biological isolates per genotype are graphed, and shade width indicates the standard deviation. (**E**) Mec1 is required for Srs2 sumoylation. Experiments were performed and data are presented as in panel A.

The online version of this article includes the following source data and figure supplement(s) for figure 5:

**Source data 1.** Original western blots for *Figure 5A, B, C and E*, indicating the relevant lanes.

**Source data 2.** Original files for western blot analysis displayed in *Figure 5A*.

**Source data 3.** Original files for western blot analysis displayed in *Figure 5B*.

**Source data 4.** Original files for western blot analysis displayed in *Figure 5C*.

**Source data 5.** Original files for western blot analysis displayed in *Figure 5E*.

*Figure 5 continued on next page*

*Figure 5 continued*

**Figure supplement 1.** Confirmation of Srs2 sumoylation.

**Figure supplement 1—source data 1.** Original western blots for *Figure 5—figure supplement 1*, indicating the relevant lanes.

**Figure supplement 1—source data 2.** Original files for western blot analysis displayed in *Figure 5E*.

reduction in Srs2 sumoylation as seen in *srs2-3KR* cells (*Figure 5A*). This result suggests that Srs2 sumoylation largely depends on Srs2 binding to PCNA, supporting the notion that the two events occur sequentially.

## Srs2 sumoylation peaks after maximal Mec1 activation and depends on Mec1

While RPA can engage with ssDNA generated in different processes, such as transcription, DNA repair, and DNA replication, only a subset of ssDNA regions is adjacent to 3'-junctions loaded with PCNA. This is because PCNA loading requires an ssDNA and dsDNA junction site with a 3' DNA end (*Moldovan et al., 2007*). We thus reasoned that Srs2 binding to PCNA may prevent Srs2 from removing RPA bound to ssDNA that is not flanked by PCNA, such as within R-loop regions and negatively supercoiling DNA regions, both of which are generated abundantly in CPT conditions due to topological stress (*Pommier et al., 2022*). This could provide a means for spatial control of Srs2's anti-RPA role. We explored whether this role could also be regulated temporally. As the genetic data described above suggest that Srs2 sumoylation acts downstream of Srs2 binding to PCNA in antagonizing RPA, we examined the timing of Srs2 sumoylation after CPT treatment.

We released G1 synchronized cells into media containing CPT and examined Srs2 sumoylation at several time points (*Figure 5B*). A similar experimental scheme was used to monitor the level of Mec1 activation using Rad9 phosphorylation as a readout (*Figure 5C*). Quantification of Srs2 sumoylated forms relative to its unmodified forms showed that its sumoylation level peaked between 1.5 and 2 hours after cells were released into CPT (*Figure 5D*). Quantification of phosphorylated forms of Rad9 relative to its unmodified form showed that checkpoint activation peaked 1 hour after cell release (*Figure 5D*). That checkpoint level peaked before Srs2 sumoylation level peaked suggests a possible dependence of the latter on the checkpoint. In testing this idea, we found that *mec1Δ* cells, which contain *sml1Δ* to support viability (*Zhao et al., 1998*), abolished Srs2 sumoylation (*Figure 5E*). Neither *sml1Δ* alone nor removal of Rad53 affected Srs2 sumoylation (*Figure 5E*), suggesting that Mec1's effect on this modification does not require the downstream Rad53 effector kinase. As Mec1 is not generally required for protein sumoylation during the DNA damage response (*Cremona et al., 2012*), its involvement in Srs2 sumoylation presents a rather unique effect.

## Examination of Mec1-S1964 phosphorylation in Srs2-mediated DDC regulation

The combined results that Srs2 sumoylation levels peak after Rad9 phosphorylation reaches maximal level and that Srs2 sumoylation depends on Mec1 raised the possibility that Mec1 hyper-activation during the late part of the DNA damage response may favor Srs2's role in DDC dampening. Recently, Mec1 protein bound to Ddc2 was shown to be auto-phosphorylated at Ser1964 in a late part of the DNA damage response after a single DNA break is induced by the HO endonuclease (*Memisoglu et al., 2019*). Further, the phosphorylation-defective mutant *mec1-S1964A* impaired checkpoint downregulation in this condition, leading to the model that Mec1-S1964 phosphorylation contributes to DDC dampening at least after HO-induced break (*Memisoglu et al., 2019*). We thus tested whether Mec1-S1964 phosphorylation is also involved in Srs2-mediated DDC regulation under CPT conditions.

First, we asked whether Mec1-S1964 phosphorylation occurs under CPT conditions using a time course scheme similar to the one used in testing Srs2 sumoylation as described above. Ddc2-bound Mec1 was recovered after immunoprecipitating Ddc2 at each timepoint and the proteins were examined by immunoblotting. To detect Mec1-S1964 phosphorylation, we used an antibody raised against the peptide containing this modification provided kindly by the Haber lab (*Memisoglu et al., 2019*). The antibody detected the Mec1 S1964 phosphorylation band in immunoprecipitated fractions only when Ddc2 was Myc tagged but not when Ddc2 was untagged (*Figure 6—figure supplement 1*).

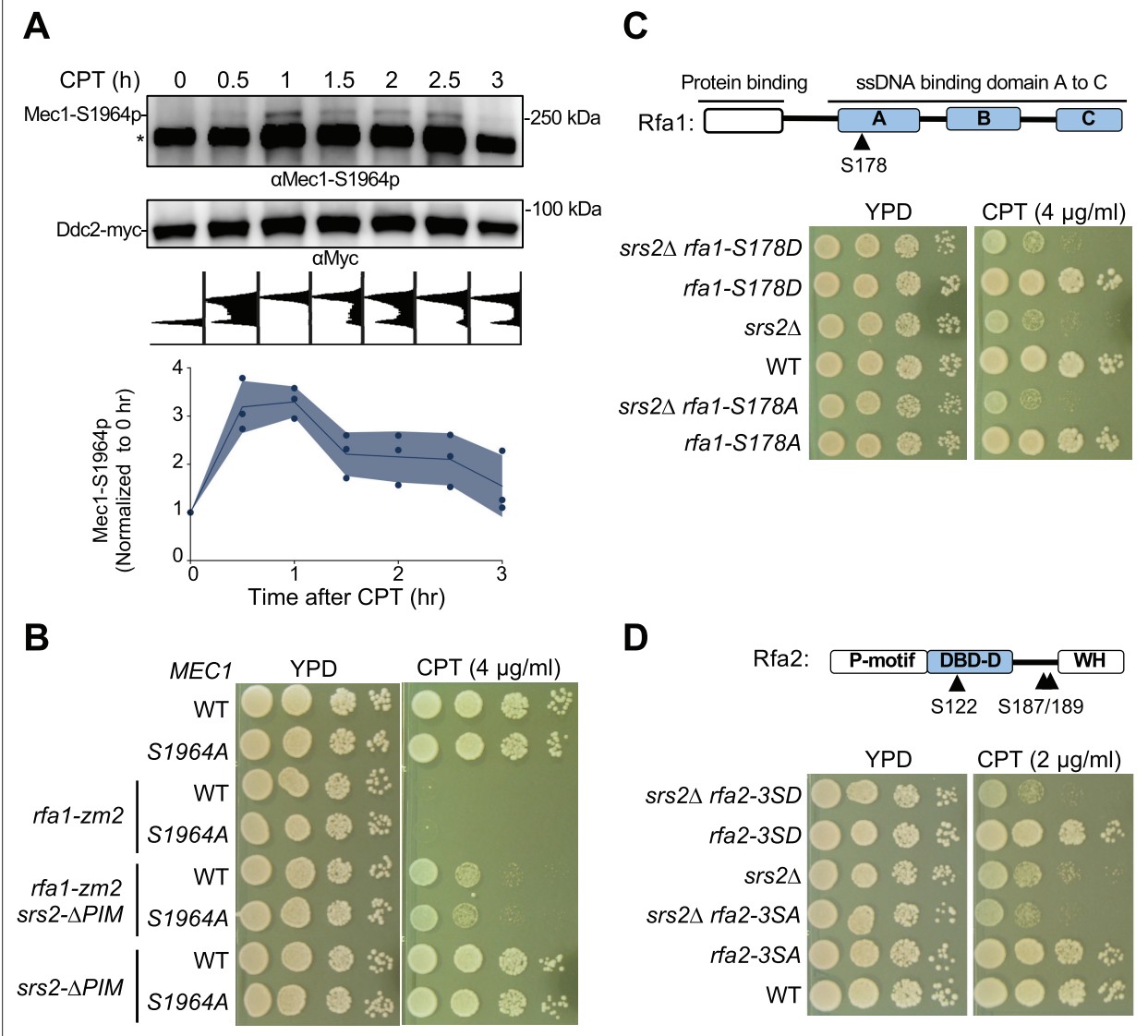

**Figure 6.** Mec1 and RPA phosphorylation did not link to Srs2 anti-checkpoint function. (**A**) Mec1 autophosphorylation at S1964 during CPT treatment. Samples were collected as described in *Figure 3B*. Ddc2-myc was immunoprecipitated and the associated Mec1 was co-purified. The anti-Mec1-S1964-p antibody detected the phosphorylated form of Mec1 and a non-specific band (*). Quantification below the blot was based on results from three biological isolates, and shade width indicates the standard deviation. (**B**) *mec1-S1964A* did not rescue *rfa1-zm2* sensitivity toward CPT. Experiments were performed as in *Figure 2C*. (**C**) Examination of the *rfa1-S178A* and *-S178D* mutants. Top: schematic of the Rfa1 protein domains and position of Ser178. ssDNA binding domain (DBD) A-C are shown. Bottom, *rfa1-S178A* and *-S178D* had no effect on *srs2Δ*'s sensitivity to CPT. Experiments were performed as in *Figure 2C*. (**D**) Examination of the *rfa2-3SA* and *-3SD* mutants. Top: schematic of the Rfa2 protein domains and position of Ser122, Ser187, and Ser189 mutated in *rfa2-3SA* and *–3SD*. ssDNA binding domain (DBD)-D is indicated. Bottom: *rfa2-3SA* has no effect on *srs2Δ*'s sensitivity toward CPT while *rfa2-3SD* showed additive effect. Experiments were performed as in *Figure 2C*.

The online version of this article includes the following source data and figure supplement(s) for figure 6:

**Source data 1.** Original western blots for *Figure 6A*, indicating the relevant lanes.

**Source data 2.** Original files for western blot analysis displayed in *Figure 6A*.

**Figure supplement 1.** Confirmation and examination of Mec1-S1964 phosphorylation.

**Figure supplement 1—source data 1.** Original western blots for *Figure 6—figure supplement 1*, indicating the relevant lanes.

**Figure supplement 1—source data 2.** Original files for western blot analysis displayed in *Figure 6—figure supplement 1*.

**Figure supplement 1—source data 3.** Original files for western blot analysis displayed in *Figure 6—figure supplement 1*.

Further, the Mec1 S1964 phosphorylation band was largely abolished in the *mec1-S1964A* cells (*Figure 6A*, *Figure 6—figure supplement 1A*). Given that the antibody also detects a closely spaced background band (*Figure 6A*, *Figure 6—figure supplement 1A*), we calculated the relative level of Mec1 S1964 phosphorylation in each sample by normalizing to this band. Quantification results showed that Mec1 S1964 phosphorylation level peaked around the same time as seen for Rad9 phosphorylation, which was 1 hour after CPT treatment (*Figure 6A*).

After confirming that Mec1 S1964 phosphorylation also took place under CPT conditions, we tested whether *mec1-S1964A* or the phosphorylation-mimetic mutant, *mec1-S1964E*, influenced *rfa1-zm2* or *srs2* growth on media containing CPT. If Mec1 S1964 phosphorylation aids Srs2-mediated RPA antagonism in CPT, we would expect *mec1-S1964A* to behave similarly to *srs2* mutants in rescuing *rfa1-zm2* CPT sensitivity, while *mec1-S1964E* should have the opposite effects. We found that *mec1-S1964A*, which showed normal CPT resistance on its own, did not affect the growth of *rfa1-zm2* cells on CPT-containing media (*Figure 6B*). *mec1-S1964A* also did not influence the growth of *srs2-ΔPIM* and *srs2-3KR* cells regardless of *rfa1-zm2* (*Figure 6B*, *Figure 6—figure supplement 1B*).

Though *mec1-S1964E* on its own did not cause CPT sensitivity, it led to slight but reproducible slow growth in cells containing *srs2-3KR* and *srs2-ΔPIM* with or without *rfa1-zm2* (*Figure 6—figure supplement 1C*). We found that neither *mec1-S1964A* nor *mec1-S1964E* affected Srs2 sumoylation levels (*Figure 6—figure supplement 1D*). Thus, Mec1, but not its S1964 phosphorylation, is required for Srs2 sumoylation. These data provide evidence that even though an HO-induced DNA break and CPT treatment both induce Mec1 S1964 phosphorylation, this modification appears to not affect Srs2-mediated DDC dampening in the latter situation.

## Genetic data suggest that known RPA phosphorylation sites do not affect DDC recovery

While we mainly focused on the identification of Srs2 features that contribute to RPA antagonism during DDC regulation, we also considered the possibility that certain regulatory RPA features may contribute to this control as well. RPA is known to be phosphorylated by Mec1 (*Chen et al., 2010*; *Faca et al., 2020*; *Lanz et al., 2021*). We thus tested whether Mec1-mediated phosphorylation may render RPA more susceptible to being removed from ssDNA by Srs2. To this end, we mutated previously identified Mec1 phosphorylation sites on the Rfa1 and Rfa2 subunits. These include Rfa1 Ser178 and three sites on Rfa2 (S122, S187, S189) (*Albuquerque et al., 2015*; *Chen et al., 2010*). We generated corresponding non-phosphorylatable mutants, *rfa1-S178A* and *rfa2-3SA* (S122A, S187A, and S189A), and phosphomimetic mutants, *rfa1-S178D* and *rfa2-3SD*.

Using genetic suppression as a readout for possible involvement of RPA mutants in Srs2-based DDC regulation, we examined the interactions between the mutants affecting RPA phosphorylation sites as described above and *srs2Δ*. If RPA phosphorylation promotes its removal by Srs2, the *SD* mutants may show suppressive interactions with *srs2Δ*, while *SA* mutants would behave like *srs2* mutants. We found that neither *rfa1-S178D* nor *S178A*, both of which showed normal CPT resistance on their own, affected *srs2Δ* cell growth on CPT-containing media (*Figure 6C*). In contrast, *rfa2-3SD* or *rfa2-3SA*, which also behaved like wild-type regarding CPT sensitivity, sensitized *srs2Δ* growth on CPT-containing media (*Figure 6D*). This negative genetic interaction contrasts with the positive interaction seen between *srs2Δ* and *rfa1-zm2* or *rfa1-t33* (*Figure 2—figure supplement 1A*; *Dhingra et al., 2021*). The differential Srs2 genetic interactions among distinct RPA alleles likely reflect the multifunctional natures of Srs2 and RPA. Collectively, these analyses suggest that the known phosphorylation sites on RPA are unlikely to be involved in Srs2-based DDC regulation.

## Discussion

Timely termination of DDC is crucial for cell survival, yet the underlying mechanisms are not well understood. Here we address this question in the context of Srs2-mediated DDC dampening in budding yeast. Our previous work reports a requirement of Srs2's ATPase activity in this role; here we examined additional Srs2 features involved in either binding to other proteins or post-translational modifications. Our results suggest that among the examined features, Srs2 binding to PCNA and its sumoylation are involved in RPA antagonism during DDC downregulation. We showed that mutants impairing these features, namely *srs2Δ-PIM* and *srs2-3KR*, suppressed both DDC abnormalities and

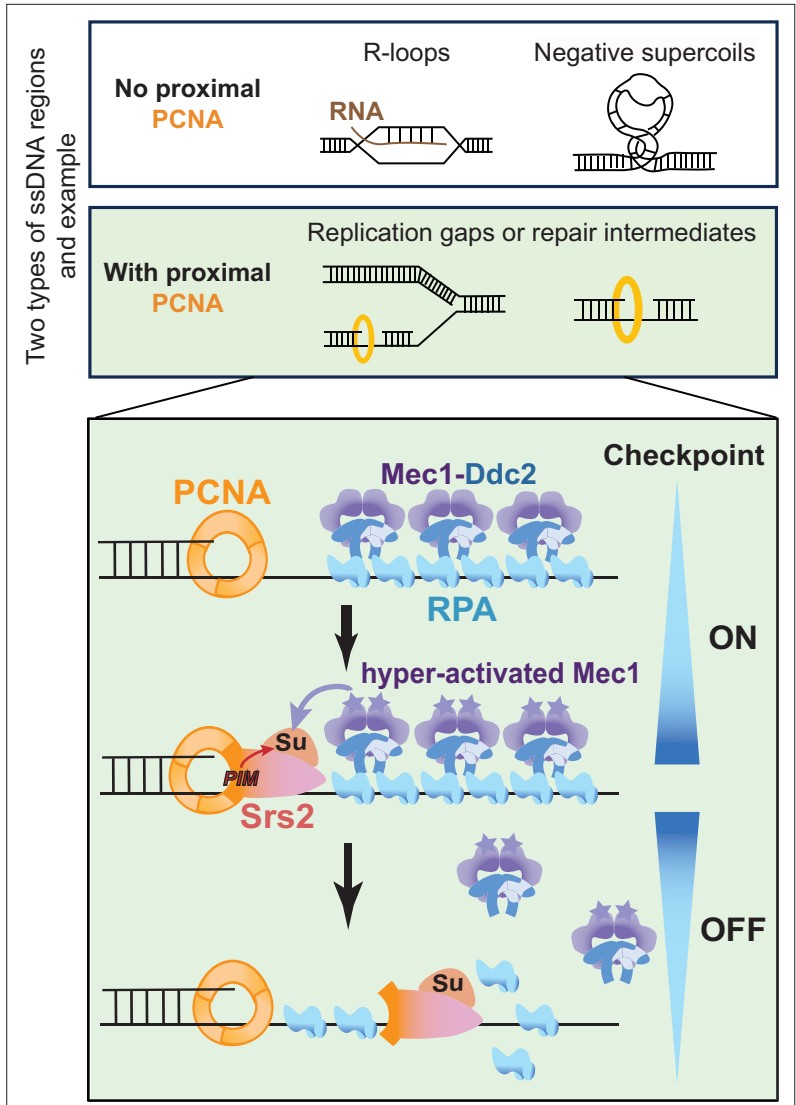

**Figure 7.** A working model for the regulation of Srs2-mediated checkpoint dampening. RPA-coated ssDNA regions generated from multiple DNA metabolic processes include those without proximal PCNA, such as ssDNA within R-loops and negatively supercoiled regions, and those with proximal PCNA, such as ssDNA gaps associated with perturbed DNA replication or repair. In latter situations, the resultant RPA-ssDNA can recruit the Mec1/Ddc2 kinase complex. Mec1 is subsequently activated by other DDC proteins, such as the 9-1-1 complex that demarcates the 5' end junctions adjacent to ssDNA regions (not drawn). After DDC is activated for a prolonged period, Srs2 recruited to PCNA that demarcates the 3' end junction flanking the ssDNA regions can be sumoylated. Srs2 sumoylation level increases as Mec1 activation heightens and can result in more efficient RPA removal, leading to Mec1 downregulation.

CPT sensitivity of *rfa1-zm2* cells. Interestingly, these features are functionally linked, since optimal Srs2 sumoylation requires its binding to PCNA. Further, the Srs2 sumoylation level peaks after Mec1 activity reaches its maximum and that Mec1 is required for this modification. We posit a model that can interpret previous and current data. In this model, Srs2 removal of RPA is favored at a subset of ssDNA-containing regions that have proximal PCNA (*Figure 7*, top). PCNA recruits Srs2 toward the adjacent RPA-ssDNA filament and promotes Srs2 sumoylation together with hyperactive Mec1 to foster downregulation of the Mec1 checkpoint (*Figure 7*, bottom).

RPA-ssDNA filaments can be separated into two groups from the perspective of PCNA loading status. The first group contains 3' DNA end at the ss-dsDNA junction that permit PCNA loading (*Figure 7*, top, green). These regions can include ssDNA gaps produced from perturbed replication

(*Sogo et al., 2002*) or from excision and repair of trapped Top1 (*Sun et al., 2020*). Due to PCNA proximity, this group of ssDNA regions can favor Srs2 recruitment and RPA removal. Since they often also contain 5' DNA ends required for loading the 9-1-1 complex (*Majka et al., 2006*), these sites can be ideal for Srs2's action in checkpoint dampening. Biochemical data suggested that PCNA is loaded exclusively at the 3' DNA ends at ssDNA-dsDNA junctions by the RFC complex in the presence of RPA, though PCNA loading could occur at 5' DNA ends in the absence of RPA (*Ellison and Stillman, 2003*; *Majka et al., 2006*). We thus do not exclude the possibility that ssDNA-dsDNA junctions containing 5' DNA ends, such as those formed at resected DNA breaks, may also allow Srs2-based RPA regulation. The second group of ssDNA regions is devoid of DNA ends and thus PCNA loading (*Figure 7*, top, white). These regions can be within R-loops or negatively supercoiled regions, which are induced by CPT treatment due to increased topological stress associated with the depletion of the functional pool of Top1 (*Pommier et al., 2022*). These regions are unlikely to contain DNA ends at ss-dsDNA junctions; thus, RPA there mainly supports DNA protection and repair and not DDC. We reason that these ssDNA sites can be shielded from Srs2 removal of RPA to allow better repair and protection. The notion that ssDNA regions close to PCNA are preferred for Srs2 anti-checkpoint action provides a rationale for how Srs2 could remove RPA at some sites while minimizing unnecessary RPA loss from other sites. Quantitative assessment of RPA residence on DNA and genome-wide mapping of Srs2 action sites in the future can further test the proposed models and provide more clarity. Regardless, results presented in this work provide evidence for the model that Srs2 can distinguish between two types of ssDNA regions using PCNA proximity as a guide for RPA removal.

Genetic analysis showed that *srs2Δ-PIM* also suppressed *rfa1-zm2* sensitivity toward MMS, though to a less degree than seen in CPT conditions. While more than one factor could contribute to this difference, we note that CPT only activates the DNA damage checkpoint, while MMS additionally induces the DNA replication checkpoint (*Menin et al., 2018*; *Redon et al., 2003*; *Tercero et al., 2003*). It is thus possible that the Srs2-RPA antagonism is more important for the DNA damage checkpoint compared with the DNA replication checkpoint. With this said, we do not exclude the possibility that the Srs2-RPA antagonism may also be relevant to specific forms of DNA repair. Further investigation of this possibility among others will shed light on differential suppressive effects seen here.

An involvement of Srs2-PCNA binding in Srs2-based RPA regulation could compensate for a lack of interaction between Srs2 and RPA. This scenario differs from Srs2-based Rad51 stripping that requires Srs2 interacts with Rad51 (*Antony et al., 2009*; *Colavito et al., 2009*; *Krejci et al., 2003*). It is thus likely that though both the anti-recombinase and anti-checkpoint roles of Srs2 benefit from selectivity of protein to be stripped from DNA, they are achieved via different means. Our previous work has provided several lines of evidence to support that Rad51 removal by Srs2 can be functionally separable from the Srs2-RPA antagonism (*Dhingra et al., 2021*). Consistent with this conclusion, deleting the Rad51 binding domain in Srs2 had no effect on *rfa1-zm2* phenotype in CPT (*Figure 2D*). This contrasts with mutating the PCNA binding and the sumoylation sites of Srs2, which suppressed *rfa1-zm2* defects (*Figure 2C*). This data provides additional evidence that Srs2 regulation of Rad51 is separable from the Srs2-RPA antagonism at genetic level. For this reason, the Srs2-based Rad51 function was not further examined in this study. However, future studies to address how Srs2 regulates RPA and Rad51 in different manners and whether there is a crosstalk between them in specific contexts will help to generate a more comprehensive understanding of both roles of Srs2.

Previous studies have suggested that both PIM and SIM aid Srs2 recruitment to sumoylated form of PCNA (*Papouli et al., 2005*; *Pfander et al., 2005*). Our data reveal that at least in CPT conditions, Srs2-PIM, but not Srs2-SIM, is involved in RPA regulation. This observation is congruent with phenotypic differences between Srs2's PIM and SIM seen in other studies and here, such as a much stronger genotoxic sensitivity of the latter (*Figure 2—figure supplement 1A*; *Fan et al., 2023*; *Kolesar et al., 2016*; *Kolesar et al., 2012*). Considering that only a small percentage of PCNA is sumoylated and mainly during S phase (*Papouli et al., 2005*; *Pfander et al., 2005*), it is probably advantageous for Srs2 not being restricted to sumoylated PCNA in order to achieve efficiency in RPA regulation.

How could Srs2 binding to PCNA helps Srs2 sumoylation? While a full answer awaits future studies, we speculate that this interaction may render Srs2 more permissive for sumoylation and/or position Srs2 in proximity with its sumoylation E3 ligase Siz2 that binds RPA (*Chung and Zhao, 2015*; *Kolesar et al., 2012*). Considering that Mec1, Ddc2, and RPA contain SUMO interaction motifs (SIMs) (*Psakhye and Jentsch, 2012*), it is possible that sumoylation could help enrich Srs2 at ssDNA-RPA sites via SIM

interactions, though other effects are possible. Compared with *srs2-3KR*, *srs2-ΔPIM* showed better suppression of *rfa1-zm2* defects. This indicates that during RPA regulation, PCNA plays additional roles besides promoting Srs2 sumoylation.

We found that Srs2 sumoylation in cells also requires Mec1, which does not promote sumoylation in general (*Cremona et al., 2012*). On the contrary, Mec1 loss causes increased sumoylation levels of multiple proteins (*Cremona et al., 2012*). It is currently unclear how Mec1 can specifically promote Srs2 sumoylation, but this may involve Mec1-mediated phosphorylation of SUMO substrates and/ or enzymes. A requirement of Mec1 for Srs2 sumoylation and the dynamic nature of sumoylation suggest that Srs2 sumoylation may be a potential timing regulator. It is possible that hyperactivation of Mec1 favors Srs2 sumoylation that may aid its own downregulation. While fully testing this idea requires time-course phosphor-proteomic studies under genotoxin treatment, we examined a Mec1 auto-phosphorylation site implicated in its downregulation after induction of a single DNA break (*Memisoglu et al., 2019*). We did not find evidence that it is involved in Mec1 dampening in CPT condition, raising the possibility that distinct mechanisms may be used in different genomic stress conditions. We also found that two phosphorylation sites on Srs2 that showed Mec1 dependency in proteomic studies had no effect in the CPT situation (*Albuquerque et al., 2015*; *Faca et al., 2020*); however, we note that these sites do not fit with the typical S/TQ Mec1 consensus sites, thus may not be the bona fide Mec1 sites. Other phosphorylation sites examined here include five Mec1 sites on Rfa1 and Rfa2 (*Albuquerque et al., 2015*; *Chen et al., 2010*). Genetic analyses did not support their roles in the Srs2 and RPA antagonism. Thus, comprehensively map DDC kinase phosphorylation sites on Srs2, RPA, and possibly their regulators, such as the RPA chaperone Rtt105 (*Li, 2018*), will be needed to identify targets of the kinases that can facilitate Srs2-based RPA regulation.

While we provide multiple lines of evidence in support of the involvement of Srs2 binding to PCNA and its sumoylation in RPA regulation during DDC downregulation, *srs2-ΔPIM* or *-3KR* alone behaved normally in checkpoint assays. A lack of defect here can be due to buffering effects, both from not-yet -characterized Srs2 features and from other checkpoint dampening pathways. In addressing the former possibility, we tested the separation pin of Srs2 (Y775).

This residue was shown to be key for the Srs2's helicase activity in vitro in a report that was published during the revision of our work (*Meir et al., 2023*). We found that *srs2-Y775A* that severely compromised the Srs2 helicase activity rescued *rfa2-zm2* sensitivity to CPT (*Appendix 1—figure 1*). This result suggests that Y775 could contribute to in vivo RPA removal; future examination of this feature in conjunction with PCNA binding and sumoylation, can extend our understanding of Srs2-mediated DDC regulation. We also provided evidence for the latter possibility since *srs2-ΔPIM* or *-3KR* showed additive phenotype when combined with a *slx4* mutant defective in DDC dampening function. The presence of multiple checkpoint-dampening factors that target different checkpoint modules highlights the importance of the process. Recent demonstration of other DNA helicases, such as the human HELB, HELQ, and BLM proteins, in stripping of RPA from ssDNA in vitro suggests that Srs2-like checkpoint downregulation may present in human cells as well (*Hormeno et al., 2022*; *Jenkins, 2021*; *Shorrocks et al., 2021*). Since RPA-ssDNA binding not only plays a key role in checkpoint, but also in many other processes, the Srs2 features uncovered here may help to better understand how RPA dynamics is controlled by helicases and other factors during additional processes beyond DDC.

# Materials and methods

**Key resources table**

| Reagent type (species) or resource | Designation | Source or reference | Identifiers | Additional information |
|---|---|---|---|---|
| Strain, strain background (*Saccharomyces cerevisiae*) | W303 | Lab stock | G16 | Strains are listed in **Supplementary file 1, table 1** |
| Sequence-based reagent | gRNA sequence | IDT | N/A | Sequences are listed in **Supplementary file 1, table 2** |
| Chemical compound, drug | Ni-NTA Agarose | QIAGEN | Cat# 30210 | |

*Continued on next page*

*Continued*

| Reagent type (species) or resource | Designation | Source or reference | Identifiers | Additional information |
|---|---|---|---|---|
| Chemical compound, drug | (S)- (+)-Camptothecin (CPT) | Sigma | Cat# C9911 | |
| Antibody | F9 anti-phosphorylated Rad53 (mouse polyclonal) | *Pellicioli et al., 2001* | N/A | WB: 1:250 |
| Antibody | Anti-Rad9 (rabbit polyclonal) | *Usui et al., 2009* | N/A | WB: 1:3000 |
| Antibody | Anti-Mec1-S1964-p (rabbit polyclonal) | *Memisoglu et al., 2019* | N/A | WB: 1:500 |
| Antibody | Anti-Srs2 (goat polyclonal) | Santa Cruz (yC-18) | Cat# sc-11991; RRID:AB_672740 | WB: 1:1000 |
| Antibody | Anti-Pgk1 (mouse monoclonal) | Invitrogen (22C5D8) | Cat# 459250; RRID:AB_2532235 | WB: 1:8000 |
| Antibody | Anti-Myc | Thermo Fisher (9E10) | Cat# 13-2500; RRID:AB_2533008 | IP: 1 µg/ml |
| Peptide, recombinant protein | α factor | GenScript | Cat# RP01002 | |
| Peptide, recombinant protein | Pronase protease | Millipore | Cat# 53702 | |
| Commercial assay or kit | 4–20% precast polyacrylamide gel | Bio-Rad | Cat# 4561096 | |
| Commercial assay or kit | NuPAGE Tris-Acetate Mini Protein Gels, 3–8%, 1.0–1.5 mm | Thermo Fisher Invitrogen | Cat# EA03752 | |
| Commercial assay or kit | NuPAGE Tris-Acetate Mini Protein Gels, 7%, 1.0–1.5 mm | Thermo Fisher Invitrogen | Cat# EA03555 | |
| Software, algorithm | Prism 9.0 | GraphPad | RRID:SCR_002798 | |
| Software, algorithm | LAS-3000 imaging system | Fuji | N/A | |
| Software, algorithm | ImageJ | FiJi | RRID:SCR_002285 | |

## Yeast strains and genetic techniques

Standard procedures were used for cell growth and media preparation. The strains used in this work are listed in *Supplementary file 1, table 1* and are isogenic to W1588-4C, a *RAD5* derivative of W303 (*Zhao and Blobel, 2005*). *srs2-3KR* was introduced using a PCR-based method (*DiCarlo et al., 2013*), while *srs2-ΔPIM*, *-SIM^{MUT}*, *-2SA*, *mec1-S1964A*, *-S1964E*, *rfa1-S178D*, *-S178A*, and *rfa2-3SA, -3SD* were generated by CRISPR-Cas9-based gene replacement as described in *Dhingra et al., 2021*. Detailed gRNA sequences were listed in *Supplementary file 1, table 2*, and repair templates for Cas9-directed mutagenesis are produced by PCR. All mutations were verified by sequencing. Standard procedures were used for tetrad analyses and genotoxin testing. All *S. cerevisiae* strains generated in this study are available upon request. For each assay, at least two random biological duplicates were used per each genotype. Cells were grown at 30°C unless otherwise stated.

## Cell synchronization and cell cycle analyses

Cells from log-phase cultures were treated with 5 µg/ml α factor (GenScript RP01002) for 1 hours, followed by an additional 2.5 µg/ml α factor for 30 minutes. When at least 95% cells were arrested in the G1 phase based on the percentage of unbudded cells, they were released into yeast extract–peptone–dextrose (YPD) media containing 100 µg/ml Protease (Millipore 53702) and 16 µg/ml CPT (Sigma C9911) for 2 hours. Cell cycle progression was monitored by standard flow cytometry analyses as described previously (*Zhao and Rothstein, 2002*). To derive the percentage of G1 cells in *Figure 3B*, the G1 region of the 'G1 sample' was used to demarcate the G1 region of the 'CPT 2 h' sample of the same strain in the same experiment.

## Protein extraction using trichloroacetic acid (TCA)

To examine the protein levels of Srs2, the phosphorylation form of Rad53, and Rad9, cell extracts were prepared as reported (*Regan-Mochrie et al., 2022*). Briefly, $2 \times 10^8$ cells were collected at indicated

time points. Cell pellets were resuspended in 20% TCA and lysed by glass bead beating. The lysate was then centrifuged to remove supernatant. Precipitated proteins were suspended in Laemmli buffer (65 mM Tris-HCl at pH 6.8, 2% SDS, 5% 2-mercaptoethanol, 10% glycerol, 0.025% bromophenol blue) with 2 M Tris added to neutralize the solution. Prior to loading, samples were boiled for 5 minutes and spun down at 14,000 rpm for 10 minutes to remove insoluble materials. Samples were separated on 4–20% Tris-glycine gels (Bio-Rad 456-1096) to examine Srs2 level and phosphorylated Rad53 form or 7% Tris-acetate gels (Thermo Fisher EA03555) to detect Rad9 phosphorylation.

## Detection of Srs2 sumoylation

Sumoylated proteins were pulled down from cells containing 8His-tagged SUMO expressed from its endogenous locus using the standard Ni-NTA pull-down method as previously described (*Ulrich and Davies, 2009*). Briefly, protein extracts prepared in 55% TCA were incubated in buffer A (6 M guanidine HCl, 100 mM sodium phosphate at pH 8.0, 10 mM Tris-HCl at pH 8.0) with rotation for 1 hours at room temperature. The cleared supernatant was obtained after centrifuging for 20 minutes and was then incubated overnight at room temperature with Ni-NTA resin (QIAGEN 30210) in the presence of 0.05% Tween-20 and 4.4 mM imidazole with rotation. Beads were washed twice with buffer A supplemented with 0.05% Tween 20 and then four times with buffer C (8 M urea, 100 nM sodium phosphate at pH 6.3, 10 mM Tris-HCl at pH 6.3) supplemented with 0.05% Tween 20. Proteins were eluted from the beads using HU buffer (8 M urea, 200 mM Tris-HCl at pH 6.8, 1 mM EDTA, 5% SDS, 0.1% bromophenol blue, 1.5% DTT, 200 mM imidazole). Samples were loaded onto NuPAGE 3–8% Tris-acetate gels (Thermo Fisher EA03752) for immunoblotting to detect both sumoylated and unmodified Srs2. Equal loading was verified using Ponceau S staining.

## Detection of Mec1 S1964 phosphorylation during CPT treatment

G1-arrested cells were filtered and resuspended in prewarmed YPD media containing 16 µg/ml CPT to allow entry into S phase. Samples were collected at indicated time points. Cells were disrupted by glass bead beating in lysis buffer, followed by centrifugation at 20,000 × $g$ for 15 minutes to obtain whole-cell extract. The lysis buffer consisted of 25 mM K-HEPES (pH 7.6), 100 mM NaCl, 100 mM K-glutamate, 5 mM Mg(OAc)$_2$, 0.02% NP40, and 0.5% Triton X-100 supplemented by protease inhibitor cocktail (Sigma P8215) and Complete Ultra EDTA-free protease inhibitor (Roche 11836145001). The activated form of Mec1 was enriched by immunoprecipitating myc-tagged Ddc2 using Protein A beads and an anti-Myc antibody (Thermo Fisher 9E10) for 2–4 hours at 4°C. Beads were washed five to six times with lysis buffer, and proteins were eluted using Laemmli buffer (65 mM Tris-HCl at pH 6.8, 2% SDS, 5% 2-mercaptoethanol, 10% glycerol, 0.025% bromophenol blue). After boiling for 5 minutes, eluted proteins were loaded onto NuPAGE 3–8% Tris-acetate gels for SDS-PAGE and subsequent immunoblotting analyses.

## Immunoblotting analysis and antibodies

After SDS-PAGE, proteins were transferred to a 0.2 µm nitrocellulose membrane (GE, #G5678144) for immunoblotting. Pgk1 was used as a loading control. Antibodies used were anti-myc (Thermo Fisher 9E10)., anti-Srs2 (Santa Cruz, yC-18), anti-Pgk1 (Invitrogen, 22C5D8), F9 (a kind gift from Marco Foiani and Daniele Piccini, The FIRC Institute of Molecular Oncology, Milan, Italy) (*Pellicioli et al., 2001*), anti-Rad9 (a kind gift from John Petrini, MSKCC, NY, USA) (*Usui et al., 2009*), and anti-Mec1-S1964-p (a kind gift from James E. Haber, Brandeis University, MA, USA) (*Memisoglu et al., 2019*). Validation of antibodies is provided either by the manufacturers or in the cited references. Membranes were scanned with a Fujifilm LAS-3000 luminescent image analyzer, which has a linear dynamic range of $10^4$. The signal intensities of non-saturated bands were measured using ImageJ software. For graphs, mean and SD are depicted.

## Statistical analyses

Pairwise comparison was conducted, and statistical differences were determined using two-sided Student's $t$-test as indicted in the graphs and figure legends using GraphPad Prism 9.0.

## Acknowledgements

We thank Drs. James Haber, Marco Foiani, Daniele Piccini, and John Petrini for sharing antibodies, and Tzippora Chwat-Edelstein for critical reading of the manuscript.

## Additional information

### Funding

| Funder | Grant reference number | Author |
|---|---|---|
| National Institute of General Medical Sciences | R35GM145260 | Xiaolan Zhao |

The funders had no role in study design, data collection and interpretation, or the decision to submit the work for publication.

### Author contributions

Jiayi Fan, Conceptualization, Data curation, Formal analysis, Investigation, Visualization, Methodology, Writing – original draft, Writing – review and editing; Nalini Dhingra, Conceptualization, Data curation, Formal analysis, Investigation, Methodology, Writing – review and editing; Tammy Yang, Vicki Yang, Data curation, Formal analysis, Investigation, Writing – review and editing; Xiaolan Zhao, Conceptualization, Resources, Formal analysis, Supervision, Funding acquisition, Investigation, Visualization, Methodology, Writing – original draft, Project administration, Writing – review and editing

### Author ORCIDs

Jiayi Fan ![ORCID] https://orcid.org/0000-0002-4317-8330
Tammy Yang ![ORCID] https://orcid.org/0009-0001-6128-3169
Xiaolan Zhao ![ORCID] https://orcid.org/0000-0002-8302-6905

Reviewer #1 (Public review): https://doi.org/10.7554/eLife.98843.4.sa1
Reviewer #2 (Public review): https://doi.org/10.7554/eLife.98843.4.sa2
Reviewer #3 (Public review): https://doi.org/10.7554/eLife.98843.4.sa3
Author response https://doi.org/10.7554/eLife.98843.4.sa4

## Additional files

### Supplementary files

Supplementary file 1. Yeast strains used in *Appendix 1—figure 1*.

MDAR checklist

### Data availability

All data generated or analysed during this study are included in the manuscript and supporting files; source data files have been provided for all gel blots displayed.

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

# Appendix 1

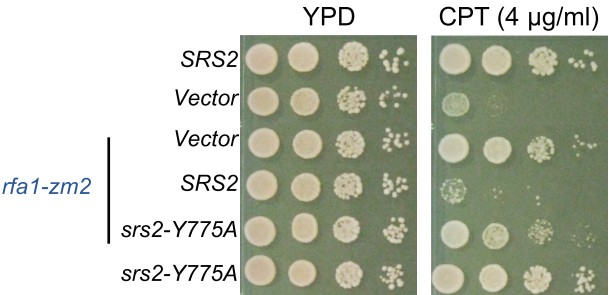

Endogenous *SRS2* deleted in all cells.

**Appendix 1—figure 1.** *srs2-Y775A* improves *rfa1-zm2* mutant growth on CPT containing media.

The endogenous *SRS2* gene was deleted in cells, while the wild-type *SRS2* or the *srs2-Y775A* allele was integrated at the *HIS3* locus (**Meir et al., 2023**). The control strain contains vector sequence integrated at the *HIS3* locus (**Meir et al., 2023**). Experiments were performed as in **Figure 2C**.

## Strains used

| Strain | Genotype | Source |
| --- | --- | --- |
| X9486-5B | *MATa his3::pRS303-p500-SRS2::HIS3 srs2Δ::KanMX* | This study |
| X9485-8D | *MATa his3::pRS303-p500::HIS3 srs2Δ::KanMX* | This study |
| X9485-8B | *MATa rfa1-zm2 his3::pRS303-p500::HIS3 srs2Δ::KanMX* | This study |
| X9486-1D | *MATa rfa1-zm2 his3::pRS303-p500-SRS2::HIS3 srs2Δ::KanMX* | This study |
| X9484-9C | *MATa rfa1-zm2 his3::pRS303-p500-Srs2-Y775A::HIS3 srs2Δ::KanMX* | This study |
| X9484-18D | *MATa his3::pRS303-p500-Srs2-Y775A::HIS3 srs2Δ::KanMX* | This study |

