## [Editor Report · eLife Assessment]

This article reports **valuable** findings on the role of the Srs2 protein in turning off the DNA damage signaling response initiated by Mec1 (human ATR) kinase. The data provide **convincing** evidence that Srs2 interaction with PCNA and ensuing SUMO modification is required for checkpoint downregulation. However, while the model that Srs2 acts at gaps after camptothecin-induced DNA damage is reasonable, direct experimental evidence for this is currently lacking. The work will be of interest to cell biologists studying genome integrity.

---

## [Referee Report · Reviewer #1 (Public review)]

Overall, the data presented in this manuscript is of good quality. Understanding how cells control RPA loading on ssDNA is crucial to understanding DNA damage responses and genome maintenance mechanisms. The authors used genetic approaches to show that disrupting PCNA binding and SUMOylation of Srs2 can rescue the CPT sensitivity of rfa1 mutants with reduced affinity for ssDNA. In addition, the authors find that SUMOylation of Srs2 depends on binding to PCNA and the presence of Mec1.

Comments on previous revisions:

I am satisfied with the revisions made by the authors, which helped clarify some points that were confusing in the initial submission.

---

## [Referee Report · Reviewer #2 (Public review)]

This is an interesting paper that delves into the post-translational modifications of the yeast Srs2 helicase and proteins with which it interacts in coping with DNA damage. The authors use mutants in some interaction domains with RPA and Srs2 to argue for a model in which there is a balance between RPA binding to ssDNA and Srs2's removal of RPA.

The manuscript mostly addresses previous concerns by doubling down on the model without providing additional direct evidence of interactions between Srs2 and PCNA, and that "precise sites of Srs2 actions in the genome remain to be determined." One additional Srs2 allele has been examined, showing some effect in combination with rfa1-zm2.

---

## [Referee Report · Reviewer #3 (Public review)]

The superfamily I 3'-5' DNA helicase Srs2 is well known for its role as an anti-recombinase, stripping Rad51 from ssDNA, as well as an anti-crossover factor, dissociating extended D-loops and favoring non-crossover outcome during recombination. In addition, Srs2 plays a key role in in ribonucleotide excision repair. Besides DNA repair defects, srs2 mutants also show a reduced recovery after DNA damage that is related to its role in downregulating the DNA damage signaling or checkpoint response. Recent work from the Zhao laboratory (PMID: 33602817) identified a role of Srs2 in downregulating the DNA damage signaling response by removing RPA from ssDNA. This manuscript reports further mechanistic insights into the signaling downregulation function of Srs2.

Using the genetic interaction with mutations in RPA1, mainly rfa1-zm2, the authors test a panel of mutations in Srs2 that affect CDK sites (srs2-7AV), potential Mec1 sites (srs2-2SA), known sumoylation sites (srs2-3KR), Rad51 binding (delta 875-902), PCNA interaction (delta 1159-1163), and SUMO interaction (srs2-SIMmut). All mutants were generated by genomic replacement and the expression level of the mutant proteins was found to be unchanged. This alleviates some concern about the use of deletion mutants compared to point mutations. Double mutant analysis identified that PCNA interaction and SUMO sites were required for the Srs2 checkpoint dampening function, at least in the context of the rfa1-zm2 mutant. There was no effect of this mutants in a RFA1 wild type background. This latter result is likely explained by the activity of the parallel pathway of checkpoint dampening mediated by Slx4, and genetic data with an Slx4 point mutation affecting Rtt107 interaction and checkpoint downregulation support this notion. Further analysis of Srs2 sumoylation showed that Srs2 sumoylation depended on PCNA interaction, suggesting sequential events of Srs2 recruitment by PCNA and subsequent sumoylation. Kinetic analysis showed that sumoylation peaks after maximal Mec1 induction by DNA damage (using the Top1 poison camptothecin (CPT)) and depended on Mec1. This data are consistent with a model that Mec1 hyperactivation is ultimately leading to signaling downregulation by Srs2 through Srs2 sumoylation. Mec1-S1964 phosphorylation, a marker for Mec1 hyperactivation and a site found to be needed for checkpoint downregulation after DSB induction, did not appear to be involved in checkpoint downregulation after CPT damage. The data are in support of the model that Mec1 hyperactivation when targeted to RPA-covered ssDNA by its Ddc2 (human ATRIP) targeting factor, favors Srs2 sumoylation after Srs2 recruitment to PCNA to disrupt the RPA-Ddc2-Mec1 signaling complex. Presumably, this allows gap filling and disappearance of long-lived ssDNA as the initiator of checkpoint signaling, although the study does not extend to this step.

Strengths:

(1) The manuscript focuses on the novel function of Srs2 to downregulate the DNA damage signaling response and provide new mechanistic insights.

(2) The conclusions that PCNA interaction and ensuing Srs2-sumoylation are involved in checkpoint downregulation are well supported by the data.

---

## [Author Response]

The following is the authors’ response to the previous reviews.

**Public Reviews:**

**Reviewer #1 (Public Review):**
Overall, the data presented in this manuscript is of good quality. Understanding how cells control RPA loading on ssDNA is crucial to understanding DNA damage responses and genome maintenance mechanisms. The authors used genetic approaches to show that disrupting PCNA binding and SUMOylation of Srs2 can rescue the CPT sensitivity of rfa1 mutants with reduced affinity for ssDNA. In addition, the authors find that SUMOylation of Srs2 depends on binding to PCNA and the presence of Mec1.Comments on revisions:I am satisfied with the revisions made by the authors, which helped clarify some points that were confusing in the initial submission.

Thank you.

**Reviewer #2 (Public Review):**
This revised manuscript mostly addresses previous concerns by doubling down on the model without providing additional direct evidence of interactions between Srs2 and PCNA, and that "precise sites of Srs2 actions in the genome remain to be determined." One additional Srs2 allele has been examined, showing some effect in combination with rfa1-zm2. Many of the conclusions are based on reasonable assumptions about the consequences of various mutations, but direct evidence of changes in Srs2 association with PNCA or other interactors is still missing. There is an assumption that a deletion of a Rad51-interacting domain or a PCNA-interacting domain have no pleiotropic effects, which may not be the case. How SLX4 might interact with Srs2 is unclear to me, again assuming that the SLX4 defect is "surgical" - removing only one of its many interactions.

Previous studies have already provided direct evidence for the interaction between Srs2 and PCNA through the Srs2’s PIM region (Armstrong et al, 2012; Papouli et al, 2005); we have added these citations in the text. Similarly. Srs2 associations with SUMO and Rad51 have also been demonstrated (Colavito et al, 2009; Kolesar et al, 2016; Kolesar et al., 2012), and these studies were cited in the text.

We did not state that a deletion of a Rad51-interacting domain or a PCNA-interacting domain have no pleiotropic effects. We only assessed whether these previously characterized mutant alleles could mimic *srs2∆* in rescuing *rfa1-zm2* defects.

We assessed the genetic interaction between *slx4-RIM* and *srs2-∆PIM* mutants, and not the physical interaction between the two proteins. As we described in the text, our rationale for this genetic test is based on that the reports that both *slx4* and *srs2* mutants impair recovery from the Mec1 induced checkpoint, thus they may affect parallel pathways of checkpoint dampening.

One point of concern is the use of t-tests without some sort of correction for multiple comparisons - in several figures. I'm quite sceptical about some of the p < 0.05 calls surviving a Bonferroni correction. Also in 4B, which comparison is **? Also, admittedly by eye, the changes in "active" Rad53 seem much greater than 5x. (also in Fig. 3, normalizing to a non-WT sample seems odd).

Claims made in this work were based only on pairwise comparison not multi-comparison. We have now made this point clearer in the graphs and in Method. As the values were compared between a wild-type strain and a specific mutant strain, or between two mutants, we believe that t-test is suitable for statistical analysis.

Figure 4B, ** indicates that the WT value is significantly different from that of the *slx4-RIM srs2-∆PIM* double mutant and from that of *srs2-∆PIM* single mutant. We have modified the graph to indicate the pair-wide comparison. The 5-fold change of active Rad53 levels was derived by comparing the values between the *srs2∆ PIM slx4RIM-TAP* double mutant and wild-type Slx4-TAP. In Figure 3, normalization to the lowest value affords better visualization. This is rather a stylish issue; we would like to maintain it as the other reviewers had no issues.

What is the WT doubling time for this strain? From the FACS it seems as if in 2 h the cells have completed more than 1 complete cell cycle. Also in 5D. Seems fast...

Wild-type W303 strain has less than 90 min doubling time as shown by many labs, and our data are consistent with this. The FACS profiles for wild-type cells shown in Figures 3C, 4C, and 5C are consistent with each other, showing that after G1 cells entered the cell cycle, they were in G2 phase at the 1-hour time points, and then a percentage of the cells exited the first cell cycle by two hours.

I have one over-arching confusion. Srs2 was shown initially to remove Rad51 from ssDNA and the suppression of some of srs2's defects by deleting rad51 made a nice, compact story, though exactly how srs2's "suppression of rad6" fit in isn't so clear (since Rad6 ties into Rad18 and into PCNA ubiquitylation and into PCNA SUMOylation). Now Srs2 is invoked to remove RPA. It seems to me that any model needs to explain how Srs2 can be doing both. I assume that if RPA and Rad51 are both removed from the same ssDNA, the ssDNA will be "trashed" as suggested by Symington's RPA depletion experiments. So building a model that accounts for selective Srs2 action at only some ssDNA regions might be enhanced by also explaining how Rad51 fits into this scheme.

While the anti-recombinase function of Srs2 was better studied, its “anti-RPA” role in checkpoint dampening was recently described by us (Dhingra et al, 2021) following the initial report by the Haber group some time ago (Vaze et al, 2002). A better understanding of this new role is required before we can generate a comprehensive picture of how Srs2 integrates the two functions (and possibly other functions). Our current work addresses this issue by providing a more detailed understanding of this new role of Srs2.

Single molecular data showed that Srs2 strips both RPA and Rad51 from ssDNA, but this effect is highly dynamic (i.e. RPA and Rad51 can rebind ssDNA after being displaced) (De Tullio et al, 2017). As such, generation of “deserted” ssDNA regions lacking RPA and Rad51 in cells can be an unlikely event. Rather, Srs2 can foster RPA and Rad51 dynamics on ssDNA. Additional studies will be needed to generate a model that integrates the anti-recombinase and the anti-RPA roles of Srs2.

As a previous reviewer has pointed out, CPT creates multiple forms of damage. Foiani showed that 4NQO would activate the Mec1/Rad53 checkpoint in G1- arrested cells, presumably because there would be singlestrand gaps but no DSBs. Whether this would be a way to look specifically at one type of damage is worth considering; but UV might be a simpler way to look. As also noted, the effects on the checkpoint and on viability are quite modest. Because it isn't clear (at least to me) why rfa1 mutants are so sensitive to CPT, it's hard for me to understand how srs2-zm2 has a modest suppressive effect: is it by changing the checkpoint response or facilitating repair or both? Or how srs2-3KR or srs2-dPIM differ from rfa1-zm2 in this respect. The authors seem to lump all these small suppressions under the rubric of "proper levels of RPA-ssDNA" but there are no assays that directly get at this. This is the biggest limitation.

CPT treatment is an ideal condition to examine how cells dampen the DNA damage checkpoint, because while most genotoxic conditions (e.g. 4NQO, MMS) induce both the DNA replication checkpoint and the DNA damage checkpoint, CPT was shown to only induced the latter (Menin et al, 2018; Minca & Kowalski, 2011; Redon et al, 2003; Tercero et al, 2003). Future studies examining 4NQO and UV conditions can further expand our understanding of checkpoint dampening in different conditions.

We have previously provided evidence to support the conclusion that *srs2* suppression of *rfa1-zm* is partly mediated by changing checkpoint levels (Dhingra et al., 2021). We cannot exclude the possibility that the suppression may also be related to changes of DNA repair; we have now added this note in the text.

Regarding direct testing RPA levels on DNA, we have previously shown that *srs2∆* increased the levels of chromatin associated Rfa1 and this is suppressed by *rfa1-zm2* (Dhingra et al., 2021). We have now included chromatin fractionation data to show that *srs2-∆PIM* also led to an increase of Rfa1 on chromatin, and this was suppressed by *rfa1-zm2* (new Fig. S2).

Srs2 has also been implicated as a helicase in dissolving "toxic joint molecules" (Elango et al. 2017). Whether this activity is changed by any of the mutants (or by mutations in Rfa1) is unclear. In their paper, Elango writes: "Rare survivors in the absence of Srs2 rely on structure-specific endonucleases, Mus81 and Yen1, that resolve toxic joint-molecules" Given the involvement of SLX4, perhaps the authors should examine the roles of structure-specific nucleases in CPT survival?

Srs2 has several roles, and its role in RPA antagonism can be genetically separated from its role in Rad51 regulation as we have shown in our previous work (Dhingra et al., 2021) and this notion is further supported by evidence presented in the current work. Srs2’s role in dissolving "toxic joint molecules” was mainly observed during BIR (Elango et al, 2017). Whether it is related to checkpoint dampening will be interesting to address in the future but is beyond of the scope of the current work that seeks to answer the question how Srs2 regulates RPA during checkpoint dampening. Similarly, determining the roles of Mus81 and Yen1 and other structural nucleases in CPT survival is a worthwhile task but it is a research topic well separated from the focus of this work.

Experiments that might clarify some of these ambiguities are proposed to be done in the future. For now, we have a number of very interesting interactions that may be understood in terms of a model that supposes discriminating among gaps and ssDNA extensions by the presence of PCNA, perhaps modified by SUMO. As noted above, it would be useful to think about the relation to Rad6.

Several studies have shown that Srs2’s functional interaction with Rad6 is based on Srs2-mediated recombination regulation (reviewed by Niu & Klein, 2017). Given that recombinational regulation by Srs2 is genetically separable from the Srs2 and RPA antagonism (Dhingra et al., 2021), we do not see a strong rationale to examine Rad6 in this work, which addresses how Srs2 regulates RPA. With this said, this study has provided basis for future studies of possible cross-talks among different Srs2-mediated pathways.

**Reviewer #3 (Public Review):**
The superfamily I 3'-5' DNA helicase Srs2 is well known for its role as an anti-recombinase, stripping Rad51 from ssDNA, as well as an anti-crossover factor, dissociating extended D-loops and favoring non-crossover outcome during recombination. In addition, Srs2 plays a key role in in ribonucleotide excision repair. Besides DNA repair defects, srs2 mutants also show a reduced recovery after DNA damage that is related to its role in downregulating the DNA damage signaling or checkpoint response. Recent work from the Zhao laboratory (PMID: 33602817) identified a role of Srs2 in downregulating the DNA damage signaling response by removing RPA from ssDNA. This manuscript reports further mechanistic insights into the signaling downregulation function of Srs2.Using the genetic interaction with mutations in RPA1, mainly rfa1-zm2, the authors test a panel of mutations in Srs2 that affect CDK sites (srs2-7AV), potential Mec1 sites (srs2-2SA), known sumoylation sites (srs2-3KR), Rad51 binding (delta 875-902), PCNA interaction (delta 1159-1163), and SUMO interaction (srs2SIMmut). All mutants were generated by genomic replacement and the expression level of the mutant proteins was found to be unchanged. This alleviates some concern about the use of deletion mutants compared to point mutations. Double mutant analysis identified that PCNA interaction and SUMO sites were required for the Srs2 checkpoint dampening function, at least in the context of the rfa1-zm2 mutant. There was no effect of this mutants in a RFA1 wild type background. This latter result is likely explained by the activity of the parallel pathway of checkpoint dampening mediated by Slx4, and genetic data with an Slx4 point mutation affecting Rtt107 interaction and checkpoint downregulation support this notion. Further analysis of Srs2 sumoylation showed that Srs2 sumoylation depended on PCNA interaction, suggesting sequential events of Srs2 recruitment by PCNA and subsequent sumoylation. Kinetic analysis showed that sumoylation peaks after maximal Mec1 induction by DNA damage (using the Top1 poison camptothecin (CPT)) and depended on Mec1. This data are consistent with a model that Mec1 hyperactivation is ultimately leading to signaling downregulation by Srs2 through Srs2 sumoylation. Mec1-S1964 phosphorylation, a marker for Mec1 hyperactivation and a site found to be needed for checkpoint downregulation after DSB induction, did not appear to be involved in checkpoint downregulation after CPT damage. The data are in support of the model that Mec1 hyperactivation when targeted to RPA-covered ssDNA by its Ddc2 (human ATRIP) targeting factor, favors Srs2 sumoylation after Srs2 recruitment to PCNA to disrupt the RPA-Ddc2-Mec1 signaling complex. Presumably, this allows gap filling and disappearance of long-lived ssDNA as the initiator of checkpoint signaling, although the study does not extend to this step.Strengths:(1) The manuscript focuses on the novel function of Srs2 to downregulate the DNA damage signaling response and provide new mechanistic insights.(2) The conclusions that PCNA interaction and ensuing Srs2-sumoylation are involved in checkpoint downregulation are well supported by the data.Weaknesses:(1) Additional mutants of interest could have been tested, such as the recently reported Pin mutant, srs2-Y775A (PMID: 38065943), and the Rad51 interaction point mutant, srs2-F891A (PMID: 31142613).(2) The use of deletion mutants for PCNA and RAD51 interaction is inferior to using specific point mutants, as done for the SUMO interaction and the sites for post-translational modifications.(3) Figure 4D and Figure 5A report data with standard deviations, which is unusual for n=2. Maybe the individual data points could be plotted with a color for each independent experiment to allow the reader to evaluate the reproducibility of the results.Comments on revisions:In this revision, the authors adequately addressed my concerns. The only issue I see remaining is the site of Srs2 action. The authors argue in favor of gaps and against R-loops and ssDNA resulting from excessive supercoiling. The authors do not discuss ssDNA resulting from processing of onesided DSBs, which are expected to result from replication run-off after CPT damage but are not expected to provide the 3'-junction for preferred PCNA loading. Can the authors exclude PCNA at the 5'-junction at a resected DSB?

We have now added a sentence stating that we cannot exclude the possibility that PCNA may be positioned at a 5’-junction, as this can be observed in vitro, albert that PCNA loading was seen exclusively at a 3’-junction in the presence of RPA (Ellison & Stillman, 2003; Majka et al, 2006).

**Recommendations For the authors:**

**Reviewer #2 (Recommendations For the authors):**
A Bonferroni correction should be made for the multiple comparisons in several figures.Specific comments:l. 41. This is a too long and confusing sentence.

Sentence shortened: “These data suggest that Srs2 recruitment to PCNA proximal ssDNA-RPA filaments followed by its sumoylation can promote checkpoint recovery, whereas Srs2 action is minimized at regions with no proximal PCNA to permit RPA-mediated ssDNA protection”.

l. 60. Identify Ddc2 and Mec1 as ATRIP and ATR.

Done.

l. 125 "fails to downregulate RPA levels on chromatin and Mec1-mediated DDC..." fails to downregulate RPA and fails to reduce Mec1-mediated DDC?

Sentence modified: “fails to downregulate both the RPA levels on chromatin and the Mec1-mediated DDC”

l. 204 "consistent with the notion that Srs2 has roles beyond RPA regulation"... What other roles? It's stripping of Rad51? Removing toxic joint molecules? Something else?

Sentence modified: “consistent with the notion that Srs2 has roles beyond RPA regulation, such as in Rad51 regulation and removing DNA joint molecules”.

l. 249 "Significantly, srs2-ΔPIM and -3KR increased the percentage of rfa1-zm2 cells transitioning into the G1 phase" No. Just back to normal. As stated in l. 258: "258 We found that srs2-ΔPIM and srs2-3KR mutants on their own behaved normally in the two DDC assays described above." All of these effects are quite small.

Sentence modified: “Compared with *rfa1-zm2* cells, *srs2-∆PIM rfa1-zm2* and *srs2-3KR rfa1-zm2* cells showed increased percentages of cells transitioning into the G1 phase”.

l. 468 "Our previous work has provided several lines of evidence to support that Rad51 removal by Srs2 is separable from the Srs2-RPA antagonism (Dhingra et al., 2021). What evidence? See my comment above about not having both proteins removed at the same time.

We have addressed this point in our initial rebuttal and some key points are summarized below. In our previous report (Dhingra et al., 2021), we provided several lines of evidence to support the conclusion that Rad51 is not relevant to the Srs2-RPA antagonism. For example, while rad51∆ rescues the hyper-recombination phenotype of *srs2∆* cells, rad51∆ did not affect the hyper-checkpoint phenotype of *srs2∆*. In contrast, *rfa1-zm1/zm2* have the opposite effects, that is, *rfa1zm1/zm2* suppressed the hyper-checkpoint, but not the hyper-recombination, phenotype of *srs2∆* cells. The differential effects of *rad51∆* and *rfa1-zm1/zm2* were also seen for the ATPase dead allele of Srs2 (*srs2K41A*). For example, *rfa1-zm2* rescued hyper-checkpoint and CPT sensitivity of *srs2-K41A* cells, while *rad51∆* had neither effect. These and other data described by Dhingra et al (2021) suggest that Srs2’s effects on checkpoint vs. recombination can be separated genetically. Consistent with our conclusion summarized above, deleting the Rad51 binding domain in Srs2 (*srs2-∆Rad51BD*) has no effect on *rfa1-zm2* phenotype in CPT (Fig. 2D). This data provides yet another evidence that Srs2 regulation of Rad51 is separable from the Srs2RPA antagonism.

l. 525 "possibility, we tested the separation pin of Srs2 (Y775), which was shown to enables its in vitro helicase activity during the revision of our work..." ?? there was helicase activity during the revision of your work? Please fix the sentence.

Sentence modified: “we tested the separation pin of Srs2 (Y775). This residue was shown to be key for the Srs2’s helicase activity in vitro in a report that was published during the revision of our work (Meir et al, 2023).”

Fig. 3. "srs2-ΔPIM and -3KR allow better G1 entry of rfa1-zm2 cells." is it better entry or less arrest at G2/M? One implies better turning off of a checkpoint, the other suggests less activation of the checkpoint.

This is a correct statement. For all strains examined in Figure 3, cells were seen in G2/M phase after 1-hour CPT treatment, suggesting proper arrest.

References:

Armstrong AA, Mohideen F, Lima CD (2012) Recognition of SUMO-modified PCNA requires tandem receptor motifs in Srs2. Nature 483: 59-63

Colavito S, Macris-Kiss M, Seong C, Gleeson O, Greene EC, Klein HL, Krejci L, Sung P (2009) Functional significance of the Rad51-Srs2 complex in Rad51 presynaptic filament disruption. Nucleic Acids Res 37: 6754-6764.

De Tullio L, Kaniecki K, Kwon Y, Crickard JB, Sung P, Greene EC (2017) Yeast Srs2 helicase promotes redistribution of single-stranded DNA-bound RPA and Rad52 in homologous recombination regulation. Cell Rep 21: 570-577

Dhingra N, Kuppa S, Wei L, Pokhrel N, Baburyan S, Meng X, Antony E, Zhao X (2021) The Srs2 helicase dampens DNA damage checkpoint by recycling RPA from chromatin. Proc Natl Acad Sci U S A 118: e2020185118

Elango R, Sheng Z, Jackson J, DeCata J, Ibrahim Y, Pham NT, Liang DH, Sakofsky CJ, Vindigni A, Lobachev KS et al (2017) Break-induced replication promotes formation of lethal joint molecules dissolved by Srs2. Nat Commun 8: 1790

Ellison V, Stillman B (2003) Biochemical characterization of DNA damage checkpoint complexes: clamp loader and clamp complexes with specificity for 5' recessed DNA. PLoS Biol 1: E33

Kolesar P, Altmannova V, Silva S, Lisby M, Krejci L (2016) Pro-recombination Role of Srs2 Protein Requires SUMO (Small Ubiquitin-like Modifier) but Is Independent of PCNA (Proliferating Cell Nuclear Antigen) Interaction. J Biol Chem 291: 7594-7607.

Kolesar P, Sarangi P, Altmannova V, Zhao X, Krejci L (2012) Dual roles of the SUMO-interacting motif in the regulation of Srs2 sumoylation. Nucleic Acids Res 40: 7831-7843.

Majka J, Binz SK, Wold MS, Burgers PM (2006) Replication protein A directs loading of the DNA damage checkpoint clamp to 5'-DNA junctions. J Biol Chem 281: 27855-27861

Meir A, Raina VB, Rivera CE, Marie L, Symington LS, Greene EC (2023) The separation pin distinguishes the pro- and anti-recombinogenic functions of *Saccharomyces cerevisiae* Srs2. Nat Commun 14: 8144

Menin L, Ursich S, Trovesi C, Zellweger R, Lopes M, Longhese MP, Clerici M (2018) Tel1/ATM prevents degradation of replication forks that reverse after Topoisomerase poisoning. EMBO Rep 19: e45535

Minca EC, Kowalski D (2011) Replication fork stalling by bulky DNA damage: localization at active origins and checkpoint modulation. Nucleic Acids Res 39: 2610-2623

Niu H, Klein HL (2017) Multifunctional roles of *Saccharomyces cerevisiae* Srs2 protein in replication, recombination and repair. FEMS Yeast Res 17: fow111

Papouli E, Chen S, Davies AA, Huttner D, Krejci L, Sung P, Ulrich HD (2005) Crosstalk between SUMO and ubiquitin on PCNA is mediated by recruitment of the helicase Srs2p. Mol Cell 19: 123-133

Redon C, Pilch DR, Rogakou EP, Orr AH, Lowndes NF, Bonner WM (2003) Yeast histone 2A serine 129 is essential for the efficient repair of checkpoint-blind DNA damage. EMBO Rep 4: 678-684

Tercero JA, Longhese MP, Diffley JFX (2003) A central role for DNA replication forks in checkpoint activation and response. Mol Cell 11: 1323-1336

Vaze MB, Pellicioli A, Lee SE, Ira G, Liberi G, Arbel-Eden A, Foiani M, Haber JE (2002) Recovery from checkpointmediated arrest after repair of a double-strand break requires Srs2 helicase. Mol Cell 10: 373-385